# "You Can Do This": Working with the Artistic Legacy of El Lissitzky

Willem Jan Renders 

Independent Researcher, 5652 EN Eindhoven, The Netherlands; w.j.renders@protonmail.com

**Abstract:** The collection of the Van Abbemuseum in Eindhoven comprises works by some 800 modern and contemporary artists, and El Lissitzky is one of the most important among them. This special position for Lissitzky is not due simply to the number of his works in the collection. In addition, his oeuvre, ideas and artistic objectives correspond closely with the museum's engagement with experimentation, radical creativity and public participation. As one of the most dynamic artists of his time, over the years Lissitzky has become more and more important to the museum. He was anything but a creator of static, self-contained works. His creativity was powerful and open to many, a mass of plans and projects bristling with life. Inspired by Lissitzky, the Van Abbemuseum was keen to make that verve and vitality tangible for today's public. The primary way to do that was to research, show and discuss his original works, but in many cases it was possible to go one step further and reconstruct what was lost or finish what the artist had started. Also, Lissitzky's works were a source of inspiration for a number of contemporary artists. In this article I will discuss how these works came to Eindhoven and give examples of how the Van Abbemuseum treated this artistic legacy in exhibitions, reconstructions, constructions and new artworks.

**Keywords:** Russian avant-garde; visual art; El Lissitzky; cultural heritage; art education; curatorial practice



## 1. Art Museums, Works of Art and the Continuation of the Creative Process

An art museum researches, collects, conserves, interprets and exhibits objects of art. How it fulfils these tasks, often remains hidden for the public. Sometimes the acquisition of a very expensive artwork gets some publicity. But, once they have entered a museum collection, the majority of artworks lead a secluded life in dark spaces with a constant temperature and humidity. The older and weaker they are, the more dangerous it becomes to put them on show. Relatively few artworks enjoy the privilege of being exhibited; they are part of a permanent or semi-permanent display in the museum. The exhibitions are what the public sees of all the work behind the scenes.

There is a way, however, in which the less lucky artworks can escape their cloisters: the temporary exhibition. When a curator finds it worthwhile to research and inventory the work of one artist, for instance, this may result in a retrospective. Many unknown works are then shown for the first time. Or it may be that an artwork can be linked to a theme. Then it may become part of a thematic exhibition. This can also be a reason to search the museum stores and temporarily free some artworks.

Both retrospectives and thematic exhibitions need more than artworks alone to tell their story to the public. Documents, photographs, videos and historical publications complete the narrative, together with accessible texts in the exhibition spaces and a catalogue for those who want more information. However, the artworks and their arrangement are the main treat; only once in a decade, or even once in a lifetime, are these works on show together.

Art exhibitions in museums treat the artwork as a unique object. And in most cases, this makes perfect sense; the artwork is the end of a unique creative process and as such



it should be respected and preserved. But sometimes it is possible or even necessary to go one step further. What can be done, for instance, if an artwork is too delicate to show? Or what if it no longer exists? Is it enough to show the few documents related to it? Or are museums allowed to go beyond that and make a reconstruction? When an artwork is unfinished, can a museum somehow 'finish' it?

In many cases exhibition makers have to come to terms with the fact that the (art)historical narrative has holes that cannot be filled. But sometimes it is possible to fill some gaps in the story of the creative process and provide a more complete picture to the public. The history of the Lissitzky collection in the Van Abbemuseum in Eindhoven offers some examples of how to take works of art as a starting point for further elaboration. This article tells the story of how a large number of works by El Lissitzky were acquired for the Van Abbe collection and how they were subsequently shown to the public[1]. In some cases, the museum's curators were able to reconstruct Lissitzky's works or add to them.

This story of collecting and exhibiting provides a look behind the scenes of museum work. And it may be the start of a critical evaluation. Did the Van Abbemuseum treat El Lissitzky's artistic heritage appropriately? Or was it not one step further but one too far?

## 2. Lissitzky Comes to Eindhoven

The Van Abbemuseum in Eindhoven (The Netherlands) owes its large collection of works by the Russian avant-garde artist El Lissitzky primarily to the efforts of its former director, Jean Leering (1934–2005). As a student of architecture, Leering became interested in the relationship between architecture and art, particularly in Constructivism, De Stijl and the Bauhaus. At a very young age, in 1964, he became director of the Van Abbemuseum[2]. Well before that time, he had discovered and studied the work of El Lissitzky (1890–1941) and he immediately began planning a major exhibition of his work, which would be the first Lissitzky retrospective in Western Europe.

Jean Leering was one of the first in Western Europe to study the works of Lissitzky thoroughly, not only by looking at and reading about them but also by making careful drawings (Figure 1). As a meticulous curator, he also listed the titles, materials, measurements, photographs and collections in a very systematic way. During preparations for his Lissitzky exhibition, this list grew into an inventory of paintings, drawings, photographs, prints and designs by Lissitzky. While working out his ideas for the exhibition, Leering also wrote to many people that had known Lissitzky: his widow Sophie Lissitzky-Küppers, who was working on her well-known monograph at the time[3], the collector and historian of literature and art Nikolai Khardziev, film maker Hans Richter, graphic designer Jan Tschichold and art collector Ella Winter.

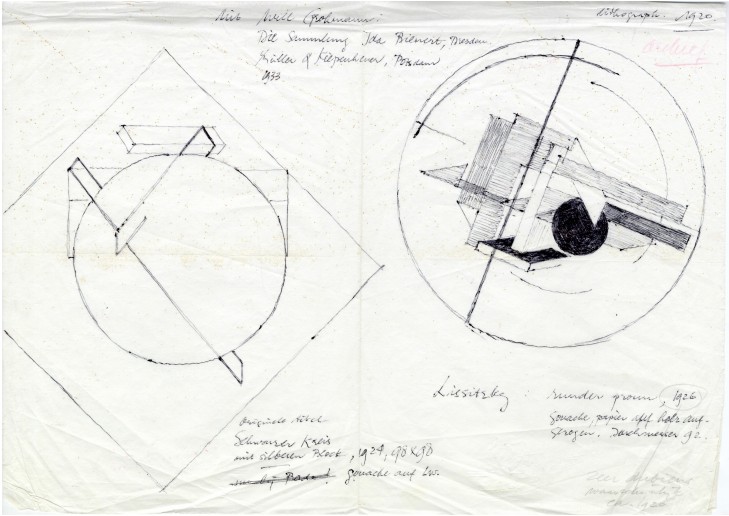

**Figure 1.** Jean Leering, one of a series of drawings after Lissitzky's 'Prouns', Van Abbemuseum exhibition archive, inv. nr. 155.

One of the people Jean Leering wrote to was Hans Klihm, a gallery owner in Munich. Leering asked him to look around for works by Lissitzky on the market. And it was this gallerist who discovered a private collection with a lot of early works by Lissitzky in Stuttgart[4]. It was owned by Ilse Vordemberge-Leda, the widow of the painter Friedrich Vordemberge-Gildewart. He had taken over Lissitzky's temporary studio in the Kestner-Gesellschaft in Hanover in 1924 after a seriously ill Lissitzky left the city at the end of 1923 for an extended period of convalescence in Switzerland. In this studio, Vordemberge-Gildewart found a portfolio with works on paper by Lissitzky. We do not know whether Lissitzky actually gave these works to Vordemberge-Gildewart, but as far as we know, he never asked for them back.

Leering requested that Ilse Vordemberge-Leda lend these newly discovered works for his exhibition. The widow agreed, but Leering had to promise that she would remain anonymous as a lender. Leering went to Stuttgart to see the works. There, a curator's dream came true. He saw a collection of over eighty unknown works by Lissitzky: early drawings, watercolours, gouaches, etchings, typographical designs and proof prints. He readily agreed to the loan conditions and also agreed right of first refusal were she to sell the works.

In the meantime, Leering kept looking for other loans. Sophie Lissitzky-Küppers asked her son, Jen Lissitzky, to go to Moscow to convince the Tretyakov Gallery to lend some of their Lissitzky works for the exhibition. She had donated almost 300 works by Lissitzky to the museum in 1959—the most important Lissitzky collection in the world—so she had the right to ask for loans. But the works in the Tretyakov Gallery could not be lent because they were 'indispensable for scientific research' (Pingen 2005, p. 224). This response illustrates the isolation and secretive treatment of the work of Lissitzky and the Russian avant-garde in general in the Soviet Union at the time.

But even without the works from Moscow, the exhibition quickly came into being as a collaboration of the Van Abbemuseum, the Kunsthalle Basel and the Kestner-Gesellschaft in Hanover. It opened in Eindhoven in December 1965, later travelling to the other venues. The essays in the catalogue, published only in German, focused on several aspects of Lissitzky's oeuvre: painting, typography, architecture, demonstration rooms and photography. The exhibition followed the arrangement of these themes. Furthermore, in the catalogue there were several key texts by Lissitzky, reminiscences by people who had known the artist and numerous illustrations in black and white. To keep the Vordemberge-Gildewart collection hidden from possible other buyers, in his foreword Leering thanked "several lenders who wish to remain anonymous", thus suggesting that the newly discovered works came from several collections (Leering et al. 1965, p. 5).

Among the works in the 1965 Lissitzky exhibition was the 'Figurinnenmappe', a portfolio of lithographs that Lissitzky published when he was working in Hanover in 1923. This was one of the first works that Leering bought when he became director of the Van Abbemuseum. Many of the drawings in the collection of Ilse Vordemberge-Leda were sketches for these lithographs. In the introductory text of this portfolio (Figure 2), Lissitzky explains that these figurines are some of the actors in the opera 'Victory over the Sun', first performed in 1913 in St. Petersburg. Malevich designed the backdrops and costumes for this first performance[5]. Lissitzky worked with Malevich and others on a new performance in Vitebsk in 1920. Inspired by this, he re-invented the opera as a mechanical performance. He designed nine puppets that would play on a stage and could be operated by one person: the 'Schaumaschinerie' ('Viewing Machine') (Figure 3).

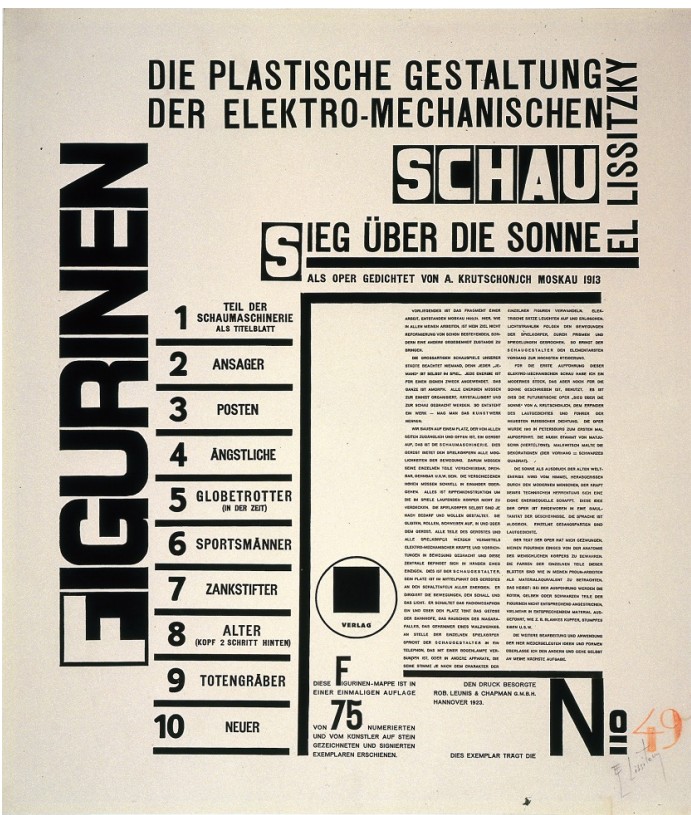

**Figure 2.** El Lissitzky, Introductory leaflet for the 'Figurinnenmappe', 1923. Van Abbemuseum. Photo: Peter Cox.

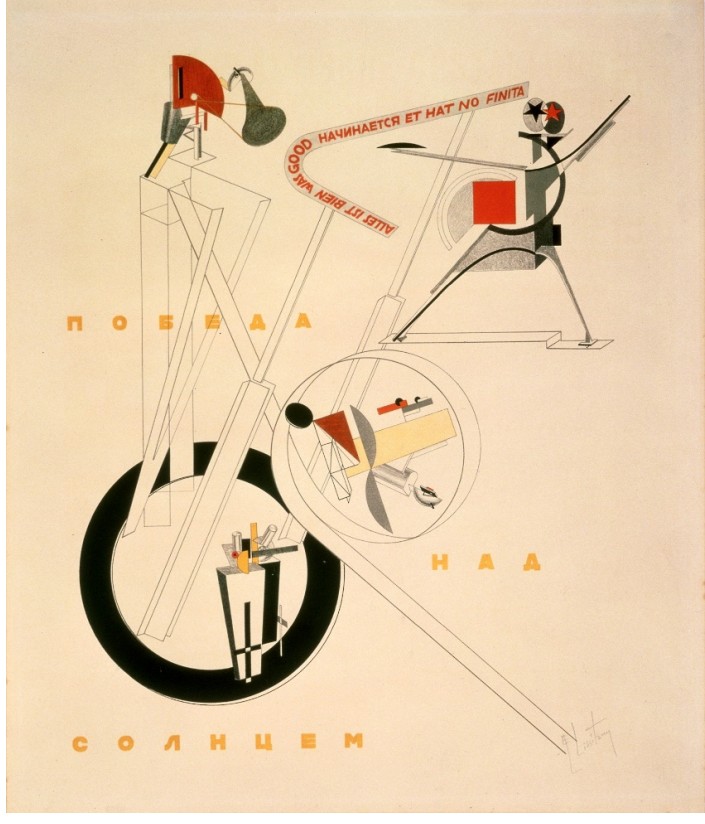

**Figure 3.** El Lissitzky, 'Schaumaschinerie' (Viewing Machine), lithograph for the 'Figurinnenmappe', 1923. Van Abbemuseum. Photo: Peter Cox.

After describing how these figures should be executed, Lissitzky ends his introduction with a statement that would be of key importance for Leering and for the treatment of Lissitzky's artistic legacy in the Van Abbemuseum:

The further elaboration and application of the ideas and forms laid down here,
I leave to others and go about my next task myself[6].

What the artist clearly states here, is that anyone is free to take the designs of the figurines and develop them further. Lissitzky encourages us to complete his project and seems to tell us: "You can do this." Mutatis mutandis, this inducement could be applied to his whole oeuvre. On several occasions, Leering cited Lissitzky's statement that his works can be a "cause for action"[7].

This incentive from the artist himself prompted the brand-new director of the Van Abbemuseum to do more with Lissitzky's work than simply study and exhibit it. Even during his training as an architect, Leering was interested in the relationship between architecture and visual art. The work of Lissitzky—also trained as an architect—was a treasure trove in this field. Leering was especially interested in the 'Prouns', a series of drawings, paintings and graphic works that Lissitzky started when teaching in 1919 at the People's Art School in Vitebsk together with Malevich.

The word 'Proun' is an abbreviation of 'Proekt Utverzhdeniia Novoga', a 'Project for the Affirmation of the New'. This series of works represents a search for new forms which, according to Lissitzky, would have their social impact and meaning in the future, not only in art but also in architecture and society[8]. It was a logical consequence that this series of two-dimensional works would eventually result in works of three dimensions. In 1923, Lissitzky designed such a space for the Große Berliner Kunstausstellung, a spatial painting as it were: the 'Prounenraum'[9].

The Prounenraum was the first actual space designed by Lissitzky. Because the original from 1923 had been lost, Leering decided to make a reconstruction of this room for his exhibition. As a model, he used a lithograph from the so-called first Kestner portfolio, a series of prints Lissitzky made in Hanover in 1923 (Figure 4).

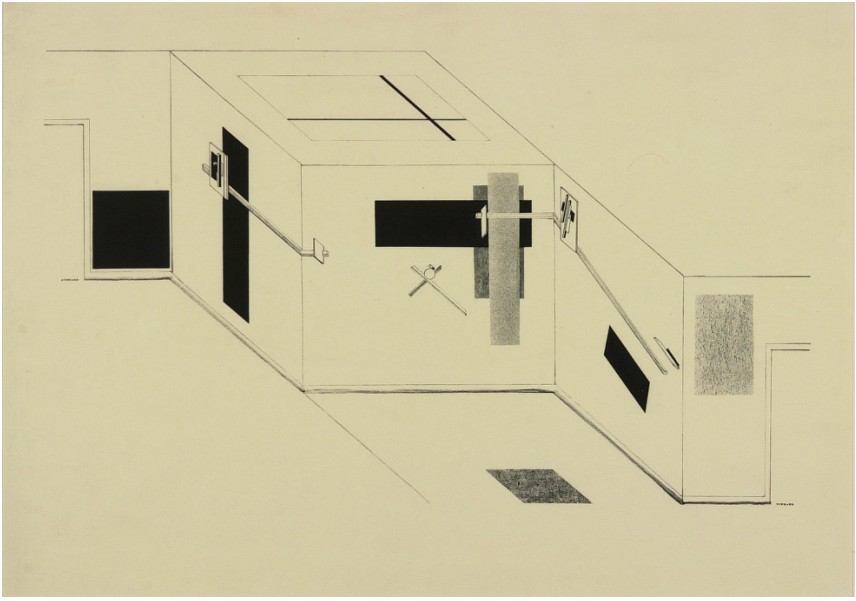

**Figure 4.** El Lissitzky, Isometric layout of the Prounenraum, lithograph for the 'Kestner Portfolio', 1923. Van Abbemuseum. Photo: Peter Cox.

This intriguing room (Figure 5) is a small space with a square floor, which can be entered through a doorway. Rectangular black and grey fields have been painted on the white walls and reliefs have been attached to these which partly continue from one wall to the next. There is a large square opening in the ceiling and cheesecloth is stretched over

the top. In this opening, two bars painted black form an asymmetrical cross. The reliefs are mainly made of wood and are composed of thin sheets, slats and bars, largely coated with transparent varnish. In addition, they contain elements which have been painted in an even black, white or grey. Here and there are narrow edges of red. One exception to the otherwise rectangular shapes is a small sphere which forms part of a relief on the back wall.

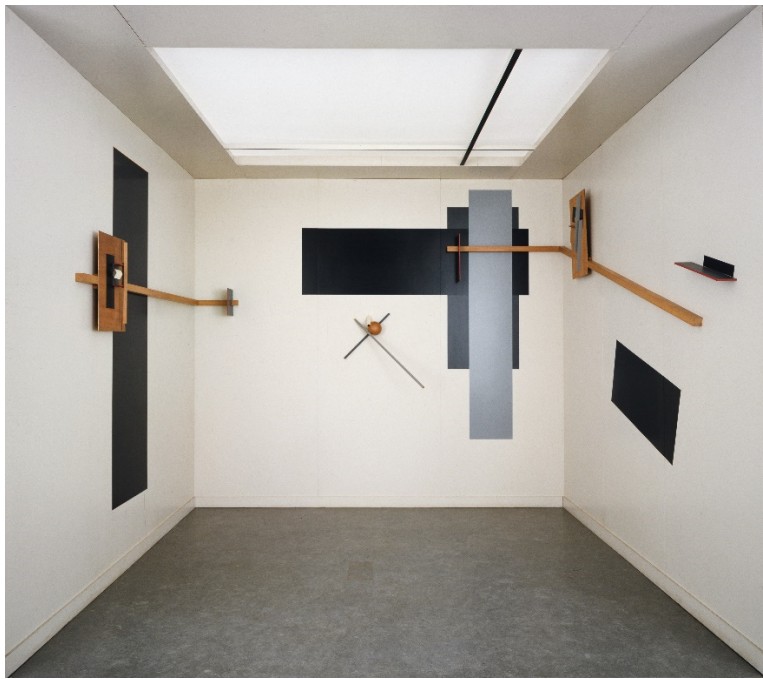

**Figure 5.** El Lissitzky, Prounenraum, 1923, reconstruction Van Abbemuseum, 1971. Van Abbemuseum. Photo: Peter Cox.

Looking back many years later, Leering described how this reconstruction came into being:

> I got to see a print of the Prounenraum that was drawn isometrically. By measuring the door height and assuming that the standard height of a door is two metres ten, I discovered that the print had been made on a scale of 1:20. Then it became clear that we could reconstruct the Prounenraum[10].

The reconstruction was meticulously executed by the technical department of the Van Abbemuseum under Leering's supervision. It was the first time since Lissitzky's death that one of his designs had been executed. This new version of the Prounenraum showed for the first time what the original might have looked like. It was one of the more eye-catching exhibits. A few years later, the museum made a new model that was more suitable for transportation. It was shown in the exhibition 'Art in Revolution' at the Hayward Gallery in London in 1971. In the interview mentioned above, Leering also made a few remarks about the purpose of such reconstructions:

> A reconstruction has a different status from an original, that should be made clear, even if it is very close to the original. Because the aim of a reconstruction is to evoke an experience almost equal to that of an original that is no longer present in its original state. In the case of reconstructions, too, I was concerned with stimulating the audience's ability to imagine[11].

All in all, the 1965 Lissitzky exhibition showed the breadth of Lissitzky's artistic production: paintings, drawings, graphic works, photographs, typographical and architectural designs (Figure 6). This retrospective took the artist out from behind the shadow of Malevich. In the catalogue introduction, Leering described chronologically the several stages of Lissitzky's multifaceted artistic production, while underlining the duality in his

oeuvre. Leering saw Lissitzky as being constantly torn between utopian ideals and concrete challenges. Against this background, he points to the importance of the Prounenraum and the other so-called demonstration rooms in Lissitzky's oeuvre:

> In my view, the design of demonstration rooms was a way of creative activity for Lissitzky in which this polarity was dissolved. Here he could create >objects< of a concrete kind, with a spiritual expressiveness that was far above that of Industrial Design. Just as with his typography, he assumed that what was shown had to be transferred to the viewers in such a way that they were included in the transfer, i.e., the creativity of the viewers themselves was set free[12].

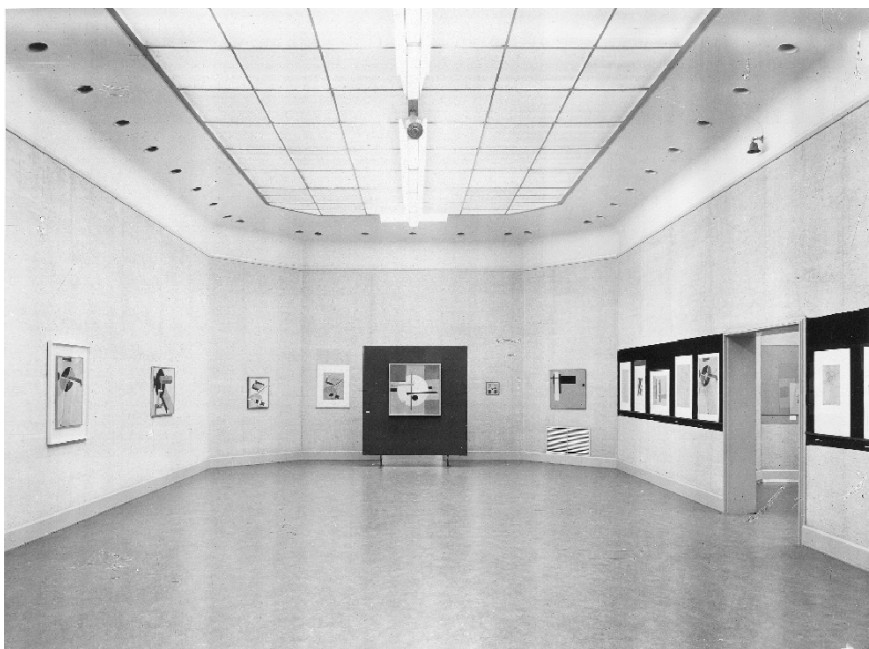

**Figure 6.** View of one of the rooms in the exhibition 'El Lissitzky', Van Abbemuseum, 1965. Photo: F. v.d. Bichelaer.

A comparison with the work of Malevich makes even more clear what Leering admired in the work of Lissitzky:

> Movement occurs in the concepts of both [artists], but with Malevich it is more as if he shows the viewer a movement in space. Lissitzky makes the viewer himself move in that space. The latter is central to Lissitzky's work, and this principle of actively involving the viewer in the image is also the starting point for his other work, such as his typography and demonstration rooms ( . . . )[13].

Leering was interested not only in the participation of the viewer but also in Lissitzky's strong social involvement. This relationship of art and society was also a special point of interest for some of the art critics writing about the exhibition. They did not see this simply as a historical aspect of Lissitzky's work, but viewed his oeuvre as inspiration for 'improving the world by integrating art and life'. The Dutch weekly magazine *De Groene Amsterdammer* wrote:

> The Eindhoven exhibition is by no means an end point: what Lissitzky had in mind is still to come. An exhibition can hardly be more inspiring[14].

Leering's inventive and innovative exhibition garnered much praise. The first time that Lissitzky was shown in Western Europe, the visionary and open character of his work was fully revealed. In retrospect, the meticulous reconstruction of the Prounenraum contributed to this appreciation. It was a necessary step to show the full extent of Lissitzky's artistic ideas.

The Vordemberge-Leda collection was one of the main attractions of the exhibition, and Leering managed to keep the lender anonymous. After the exhibition tour, this collection came to the Van Abbemuseum on long-term loan. Although the director still had first option to purchase it, finding the money was by no means an easy task.

Successful museum exhibitions of an artist's work often have a positive effect on the price, especially if there are few works on the market. In retrospect, it would have been better to purchase the Lissitzky collection before the exhibition opened. But for a municipal museum a large purchase requires a great deal of consultation, and such a decision could not be made during the short period spent organising the exhibition.

The opportunity to acquire this collection was certainly unique. There were not many works by Lissitzky in the Netherlands at the time. The Gemeentemuseum in The Hague had one painting, 'Proun GBA' (1923). The Stedelijk Museum in Amsterdam had the Proun portfolio, the Kestner portfolio and the Figurinnen portfolio—only graphic works. In some private collections (Oud, Alma en Zwart) there were typographical and architectural designs.

The acquisitions committee was enthusiastic when Leering proposed buying the Lissitzky collection. They immediately saw the importance of the collection for the Van Abbemuseum: as Amsterdam had Malevich and The Hague had Mondrian, Eindhoven would have Lissitzky. Moreover, this collection showed the multifaceted nature of Lissitzky's oeuvre. But after the exhibition tour the prices of Lissitzky's work rose sharply, and the asking price for the Vordemberge-Leda collection rose to more than double the original amount: from the insured value of 172,000 guilders to 360,000 guilders.

The exhibition was not the only reason for the change in price. In violation of the agreement, the art dealer Klihm had also offered the collection to another buyer. However, the new price was still considered reasonable, and Leering promised the widow to complete the deal before 1 May 1968. In the meantime, he tried to persuade the state to buy the collection, but the Dutch Minister of Culture declined. Further efforts to obtain private funding were also unsuccessful.

After all attempts failed, in desperation Leering wrote a press release on 29 April 1968 with the headline "Works of El Lissitzky lost to the Netherlands". This had the golden touch. Several articles appeared in Dutch newspapers and soon a large Dutch bank announced that it was willing to lend the purchase price. The minister agreed to the loan and after a long and tumultuous meeting on 27 May 1968, the city council of Eindhoven also agreed. The Lissitzky collection finally entered the Van Abbemuseum and became the specialty of the house[15]. After the Tretyakov Gallery, it is the largest collection of works by this artist. In the years to come, these works would prove a rich source of inspiration, not only for art historians and art lovers but also for a broader public and for contemporary artists.

Immediately after Lissitzky's work became the focus of the museum's collection, everything to do with this artist began to be collected: books and magazines that Lissitzky had designed or to which he had contributed in other ways, texts that he had written, photographs, letters and other correspondence. In 1970, for example, Jean Leering managed to purchase one of the numbered copies of 'The Story of Two Squares' (Figure 7), as well as the booklet 'Kunstismen' (Figure 8). Reprints of works by Lissitzky were also acquired, including the famous poster 'The Red Wedge Against the Whites' (Figure 9). In addition, all publications about Lissitzky were collected in the museum's library. In the years following Jean Leering's directorship, the Van Abbemuseum grew into a place where Lissitzky's work was documented and studied. This generated many international contacts and offered opportunities for cooperation on new Lissitzky projects.

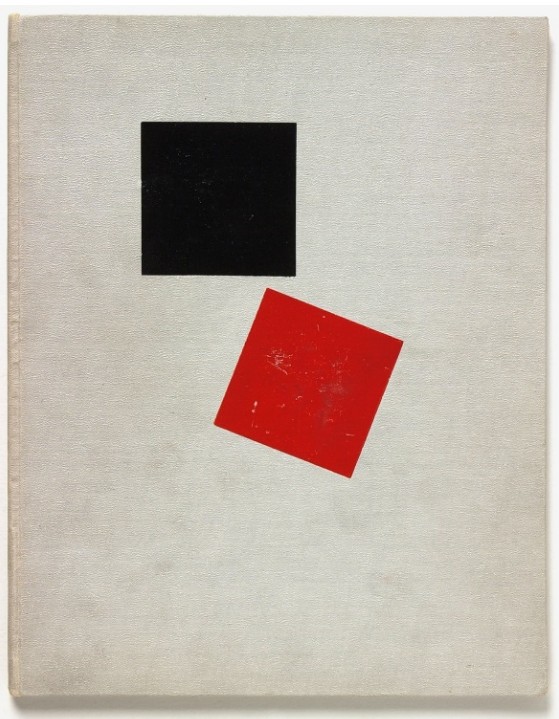

**Figure 7.** Kazimir Malevich and El Lissitzky, 'The Story of Two Squares', Berlin, 1922 (Lissitzky and Malevich 1922). Van Abbemuseum. Photo: Peter Cox.

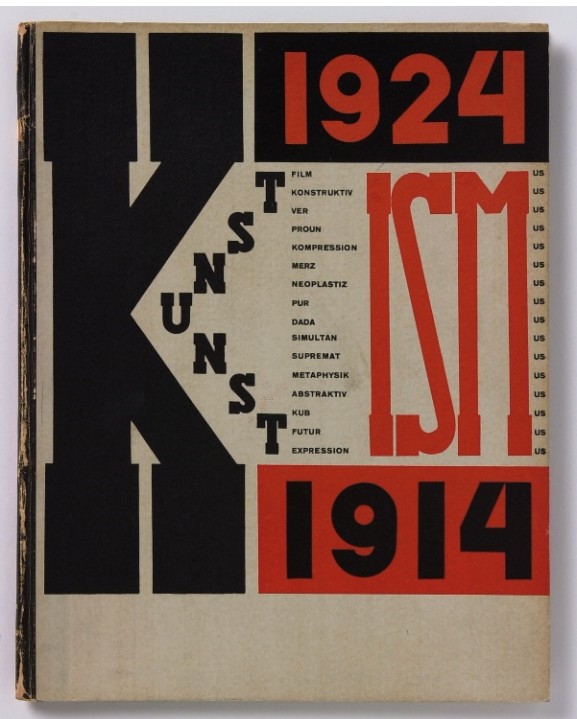

**Figure 8.** El Lissitzky and Hans Arp, 'Kunstismen 1914–1924', Zürich, 1925 (Lissitzky and Arp 1925). Van Abbemuseum. Photo: Peter Cox.

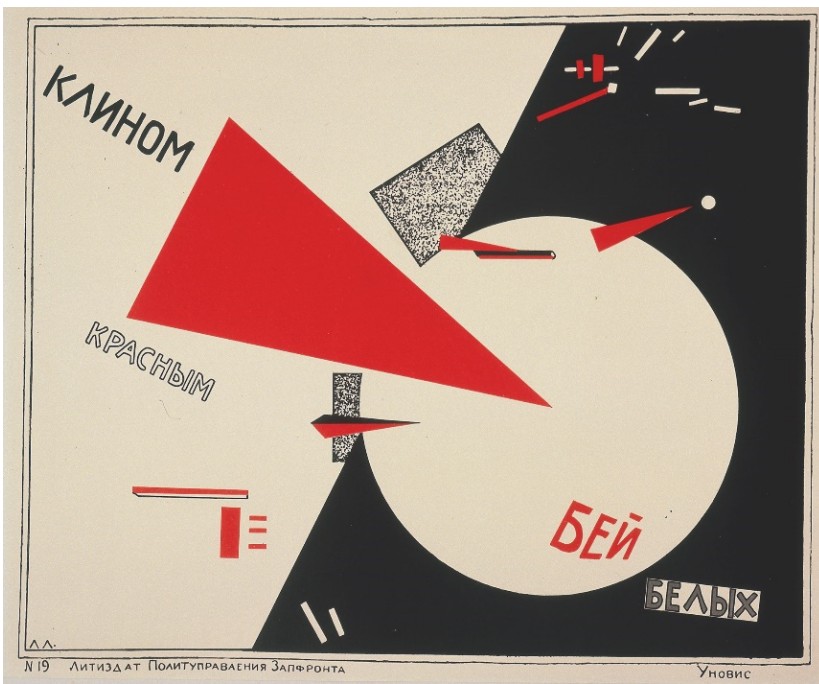

**Figure 9.** El Lissitzky, 'The Red Wedge Against the Whites', 1920, reprint 1966.

## 3. A Major Retrospective

What was still impossible for director Jean Leering in 1965 was realised 25 years later by one of his successors. In 1990, director Jan Debbaut together with the curators Mariëlle Soons, Caroline de Bie and Frank Lubbers, organised a major Lissitzky exhibition in collaboration with the Tretyakov Gallery in Moscow. By combining the two largest Lissitzky collections with many other important loans, it was possible to produce an enormous retrospective to mark the hundredth anniversary of Lissitzky's birth. A total of 330 works were shown, 250 of which came from the Moscow and Eindhoven collections. At last, the full breadth of Lissitzky's oeuvre could be admired: early travelling sketches, book illustrations, Prouns, Figurines, typographical designs, photography, demonstration rooms, architecture and exhibition designs.

The exhibition started in Moscow in 1990 and then travelled to the Van Abbemuseum, where it opened in December. In the following year the exhibition was also shown in the Fundaçion Caja de Pensiones in Madrid and in the Musée d'art moderne de la ville de Paris[16]. It was accompanied by a richly illustrated catalogue in Dutch, English, French and Spanish for the European tour (Debbaut 1990). Several international Lissitzky experts contributed: Yve-Alain Bois, S.O. Khan-Magomedov, Kai-Uwe Hemken, Jean Leering, Peter Nisbet and M.A. Nemirovskaya. Thus, Lissitzky's multifaceted oeuvre was analyzed from many sides and the catalogue provided an up-to-date overview of the state of research in the field.

Also included in the catalogue was a bitter comment by Lissitzky's son Jen, who spoke at the opening of the exhibition (Figure 10). He did not agree with M.A. Nemirovskaya's article about the Lissitzky collection in the Tretyakov Gallery. Jen Lissitzky argued that his mother, Sophie Lissitzky-Küppers, had been forced to sell her husband's work to the Moscow museum in 1958 for a paltry sum, and that it had certainly not been his father's intention that this collection should end up there. A few years later, it turned out that this commentary was a prelude to a legal claim: in 1992, Jen Lissitzky demanded the works from the former Vordemberge-Leda collection that had been bought by the Van Abbemuseum. However, he did not pursue this case in court[17].

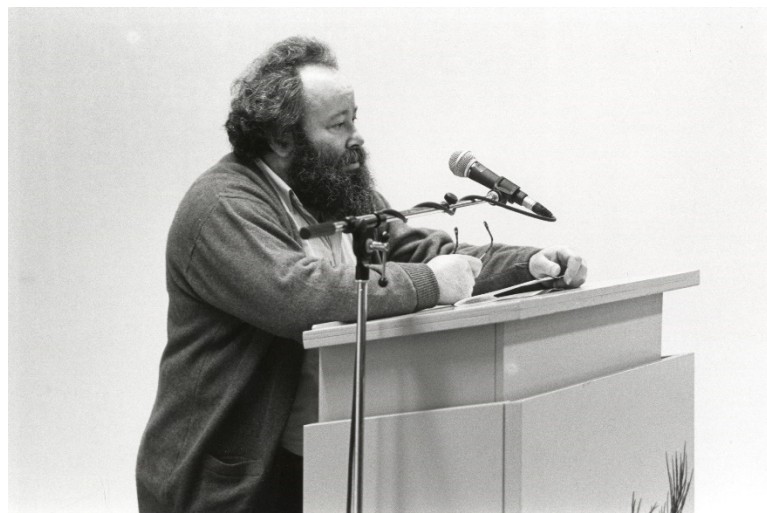

**Figure 10.** Jen Lissitzky speaking at the opening of the exhibition 'El Lissitzky', Van Abbemuseum, 15 December 1990. Photo from Van Abbemuseum exhibition archive. Photographer unknown.

For this major retrospective, part of another of Lissitzky's demonstration rooms was reconstructed: the 'Abstract Cabinet'. Lissitzky designed this room in 1927 for Alexander Dörner, then director of the Provincial Museum in Hanover. Dedicated to the presentation of contemporary art, this exhibition room also had its own artistic value. The original was destroyed by the Nazis in 1937 and the museum in Hanover made its first reconstruction in 1968, followed by several other versions[18]. The reconstruction in the Van Abbemuseum consisted of two of the walls of this room, both with the black and white slats that formed a dynamic background for the paintings exhibited on them. One of the walls had the slideable panels in which graphic works could be shown (Figure 11).

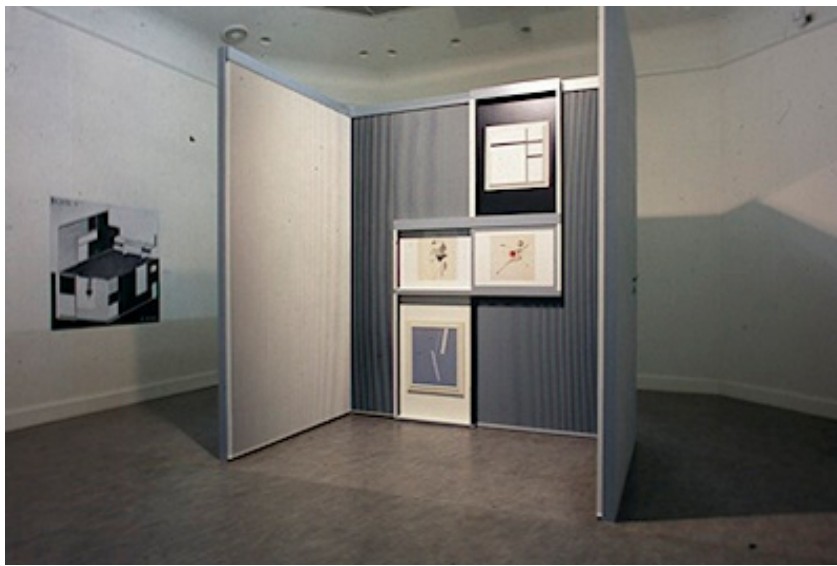

**Figure 11.** El Lissitzky, One wall of the 'Abstract Cabinet', 1927 (reconstruction 1990). Collection Van Abbemuseum. Photo from Van Abbemuseum exhibition archive. Photographer unknown.

If we compare the Eindhoven 'Abstract Cabinet' with the one in Hanover, we notice that the German version is more elaborate and precise. Looking back, I wonder why there was no cooperation between the museums to develop this reconstruction together. Surely, the Van Abbemuseum would have benefited from this.

## 4. One of the Few Proun Paintings

Although the collection in the Van Abbemuseum provided a good picture of the breadth and variety of Lissitzky's oeuvre, it still lacked a painting by the artist: one of his Prouns. Since only a few of these paintings have been preserved—about 25 in total—and because a large number of these works are in public collections, Prouns rarely come onto the market. And if they are offered for sale, they are virtually unaffordable for a smaller museum. It is therefore a miracle that the Van Abbemuseum, in the person of director Jan Debbaut, succeeded in 1997 in purchasing the painting 'Proun P23, no. 6' (Figures 12 and 13) from the estate of collector and art dealer Eric Estorick.

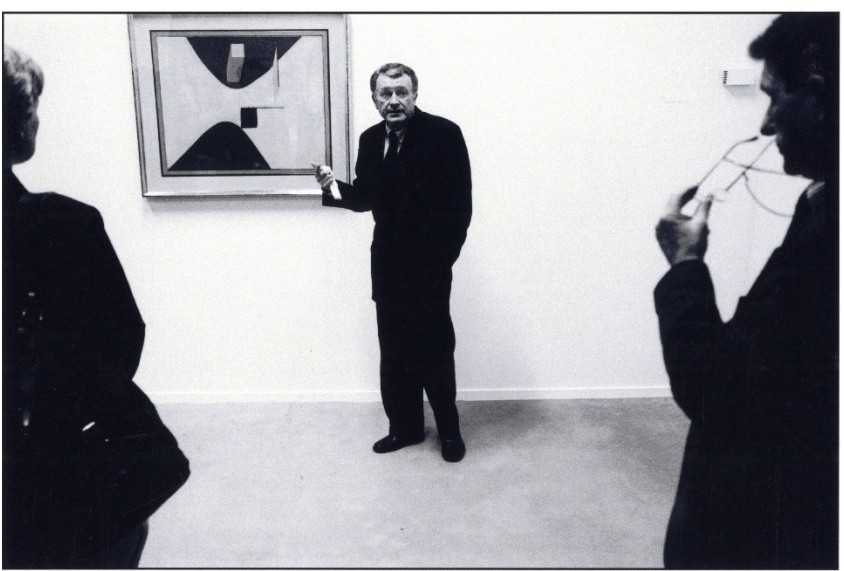

**Figure 12.** Director Jan Debbaut presents the newly acquired 'Proun P23, no. 6', 8 April 1997. Photo from Van Abbemuseum exhibition archive. Photo: Joep Lennarts.

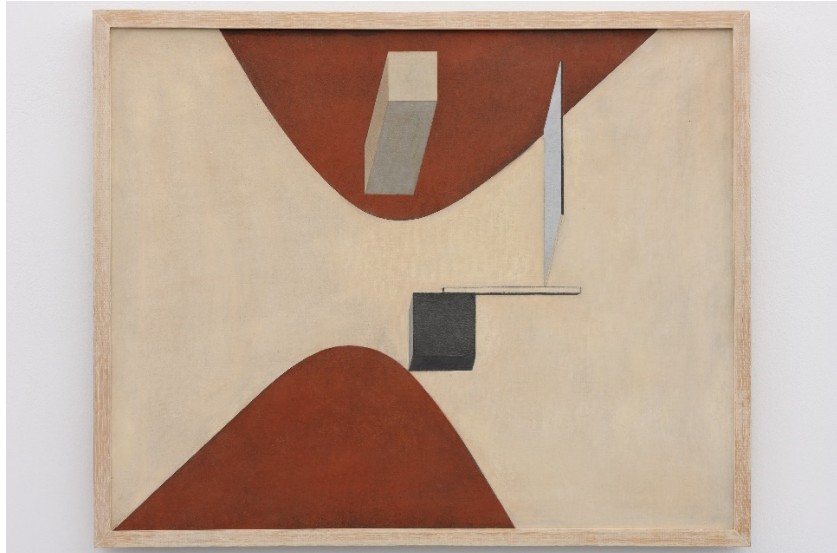

**Figure 13.** El Lissitzky, 'Proun P23, no. 6', 1919 (?). Van Abbemuseum. Photo: Peter Cox.

The painting had already been shown twice at the Van Abbemuseum, in the 1965 and 1990 exhibitions. Jan Debbaut was one of the first to contact the Estorick heirs and, thanks to smart negotiation, managed to keep his bid on the table. He found six financing partners in the Netherlands. In the meantime, the price rose dangerously high due to exchange rate fluctuations. In addition, other interested parties came forward. The sale was finally

concluded on 20 February 1997 for the amount of $1,414,000 and this Proun painting was added to the Lissitzky collection of the Van Abbemuseum (Pingen 2005, p. 484).

## 5. Lissitzky and Contemporary Art

In 2006, the Van Abbemuseum asked the artist Deimantes Narkevicius (1964, Uthena, Lithuania) to curate a 'Plug In' presentation, one in a series of interventions in the then semi-permanent presentation of the museum's collection[19]. The result, 'Plug In #6', was curated by director Charles Esche and curator Christiane Berndes. It consisted of an ambulatory space in which a selection of graphic works and drawings by Lissitzky was shown and a central projection room that showed Narkevicius's work 'Energy Lithuania' (2006) (Figures 14 and 15).

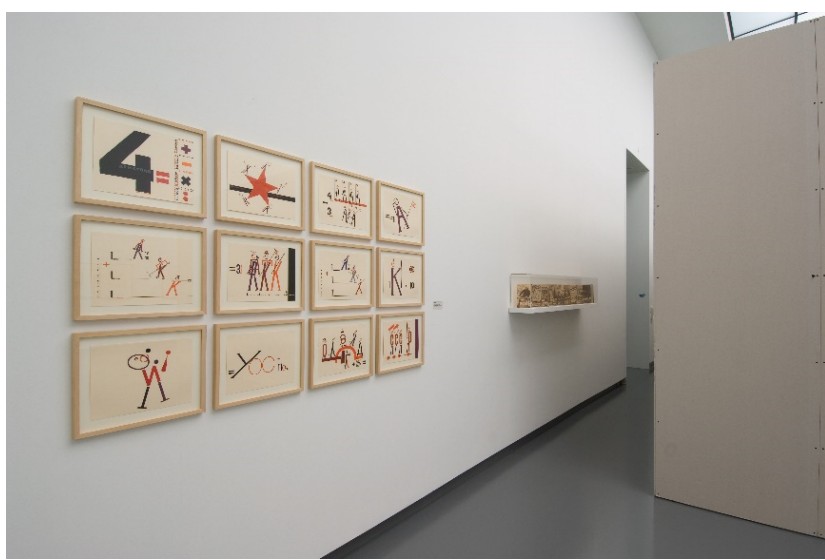

**Figure 14.** View of some of the works in the corridor of the presentation 'Plug In #06: El Lissitzky, Deimantas Narkevicius', Van Abbemuseum, 2006. Front: El Lissitzky, 'Die vier Grundrechnungsarten' ('The Four Basic Calculation Types'), 1928 (reprint 1976). Collection Van Abbemuseum. Back: El Lissitzky, 'Catalogue of the Pressa Exhibition', Cologne 1928. Van Abbemuseum. Photo: Peter Cox.

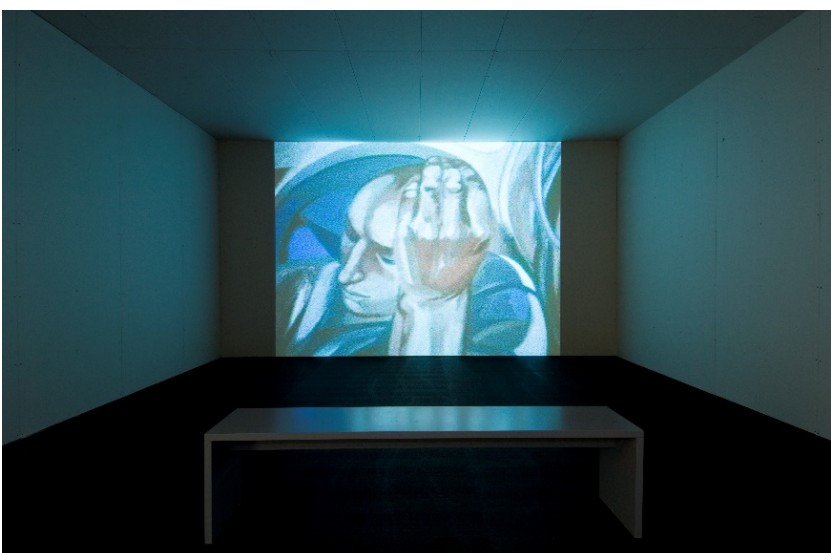

**Figure 15.** Room view of the presentation 'Plug In #06: El Lissitzky, Deimantas Narkevicius', Van Abbemuseum 2006. Still from the film 'Energy Lithuania' by Deimantas Narkevicius, 2006. Photo: Peter Cox.

Like many of the works of Narkevicius, the film 'Energy Lithuania' is a cinematic essay on the Soviet period in his homeland. It tells the story of 'Elektronai', a Lithuanian town around a former Soviet power plant. It includes impressive images of the exterior and interior of the industrial building—still in use at the time—and interviews with people who used to work there. With its huge socialist realist tableaus and clear architectural references to the Communist period of the recent past, the power plant became a kind of museum after the collapse of the Soviet Union.

The room surrounding the projection room showed a broad selection by Narkevicius of works from the museum's Lissitzky collection. There were some of the sketches and lithographs for the 'Figurinnenmappe', discussed above. Also shown were the prints of 'Die vier Grundrechnungsarten', a proposal for the combined education of basic mathematics and the structure of Soviet society (1928, reprint 1976). There was the catalogue for the Pressa exhibition in Cologne with the famous leporello showing the huge propagandistic photomontages (1928). Furthermore, there were some of the drawings and lithographs for the 'Proun Portfolio', the first series of prints for the Kestner Foundation in Hanover made in 1923. And to conclude there was a reading table with, among other works, *The Story of Two Squares*, the children's book Lissitzky made together with Malevich in 1922 (Figure 7) and the booklet *Kunstismen*, the catalogue of art movements that Lissitzky compiled together with Hans Arp in 1924 (Figure 8).

In this context these silent works mainly in black, white and red, containing the hopes of a new era, formed a contrast with the moving images in technicolour of the film on the Soviet power plant inside the screening room. Looking back on this project, Narkevicius says:

> The total presentation had something of a stylised cinema interior, with an entrance showing posters and a place to sit and drink before entering the 'inner sanctum' where the dream is projected. ( . . . ) In this presentation the 'film posters' in the 'entrance' were all works by Lissitzky connected with the propaganda of the early Soviet state and its hopes for mechanisation and industrialisation. The 'cinema' showed a striking example of one of the dystopian places where this utopian ideology came to an end: the Lithuanian power plant. The studies for 'Victory over the Sun' for instance, formed an interesting contrast in this context. In many ways you could compare Soviet history at its beginning and end. ( . . . ) In the oeuvre of Lissitzky I like the fact that he was an international artist looking for common ground for the new art of East and West. In his creative work he was both a visionary and spiritual protagonist and a very practical person involved in all kinds of projects dedicated to changing society. This is a combination that one rarely finds in artists. In that sense he is still of value for us today.

## 6. Taking Lissitzky's Designs One Step Further

The idea to create a new series of Lissitzky exhibitions in the Van Abbemuseum dates from 2008. Director Charles Esche saw Lissitzky as an important precursor of many contemporary artists, someone who was aware of his social and political role as well as of his demands regarding artistic innovation. Because of that, Esche wanted to give the Lissitzky collection a central place in the museum. The aim of this new exhibition series was to show Lissitzky's oeuvre, as well as work by his historical and contemporary colleagues in several different contexts. This would cast new light on his oeuvre and the context in which it was produced, as well as on Lissitzky as a person.

The first exhibition in the series was 'Lissitzky+'. It was based on the 1913 Russian futurist opera 'Victory over the Sun', mentioned above. Inspired by the 1920 Vitebsk performance of this opera, Lissitzky designed a dynamic stage and a number of 'figurines', doll-like figures. The collection acquired by Jean Leering included some sketches for these figurines, as well as a number of printer's proofs and the ultimate result, the portfolio of lithographs.

As we have seen, Lissitzky's introduction to the 'Figurinenmappe' (Figure 2) had inspired director Jean Leering in 1965. But this same text continued to inspire his successor in 2008. Not only are Lissitzky's ideas on theatre, as expressed here, very original, in this introduction we can also read about his intentions with this graphic project and its possible continuation. As Leering noted, the last sentence of the introduction makes it clear that Lissitzky actually provides instructions for anyone who would like to create these figurines based on these illustrations. But strangely enough nobody had ever actually done this.

Encouraged by Lissitzky's advice, Charles Esche, together with guest curator Professor John Milner and me, seized the initiative to construct these figurines. We even went one step further: thinking along these lines, other sketches and designs by Lissitzky could be developed into three dimensions as well. Thus, the concept of the transformation of flat designs into spatial ones became the underlying principle for the whole exhibition 'Lissitzky+'. The newly developed models were presented in different rooms, often together with their two-dimensional counterparts.

Carefully looking at Lissitzky's drawings and lithographs in the museum's collection and studying the other documentation of the figurines, John Milner and his son, modelmaker Henry Milner, constructed four models: the 'Announcer', the 'Time Traveler', the 'Gravediggers' and the 'New Man'. Moreover, in the pond of the museum an eight-metre-high statue of the 'Gravediggers' arose (Figures 16–23).

In the abovementioned introduction to the portfolio, Lissitzky also describes what he calls the 'Schaumaschinerie', the 'Viewing Machine'. It is the center of the play: a flexible construction of ribs that can be viewed from every side. In it we can see the so-called 'Playing Bodies' ('Spielkörper'), mechanical puppets that take the place of players. For the exhibition 'Lissitzky+', this complex design was also transformed into three dimensions (Figure 24).

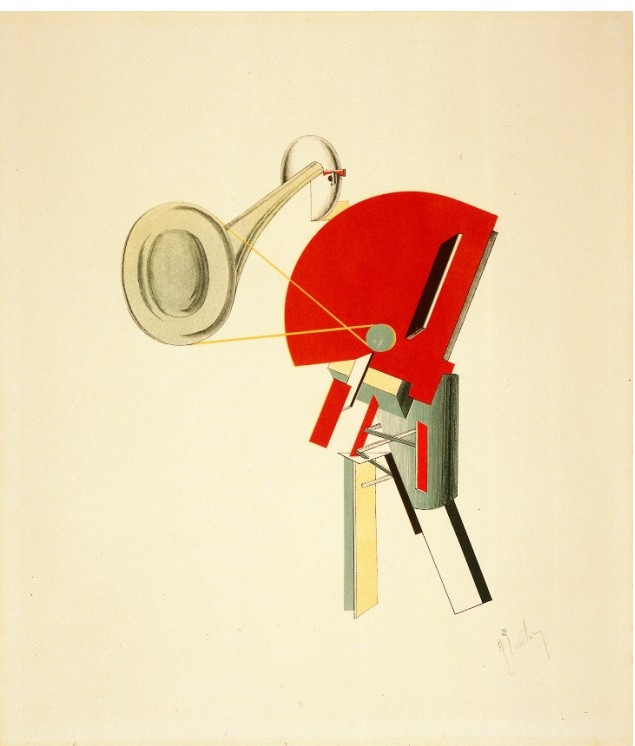

**Figure 16.** El Lissitzky, 'The Announcer', lithograph for the 'Figurinnenmappe', 1923. Van Abbemuseum. Photo: Peter Cox.

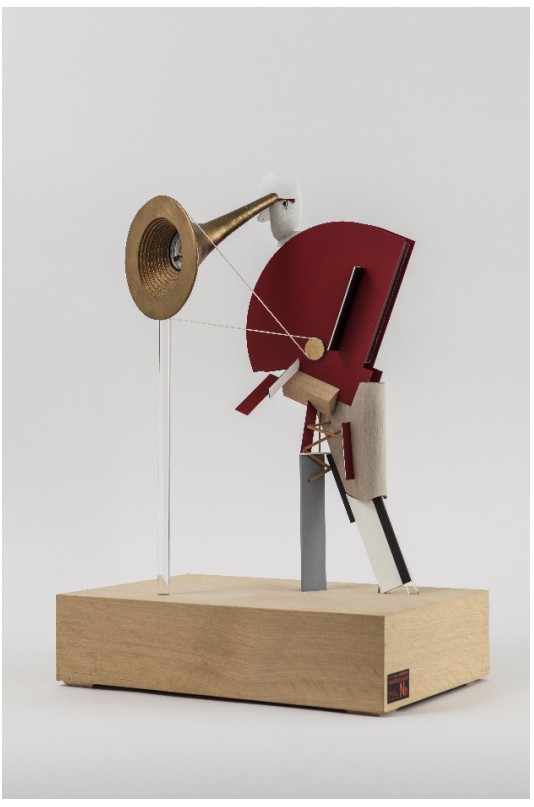

**Figure 17.** Model after El Lissitzky, 'The Announcer', 1923, construction by Henry Milner, 2009. Van Abbemuseum, photo Perry van Duijnhoven.

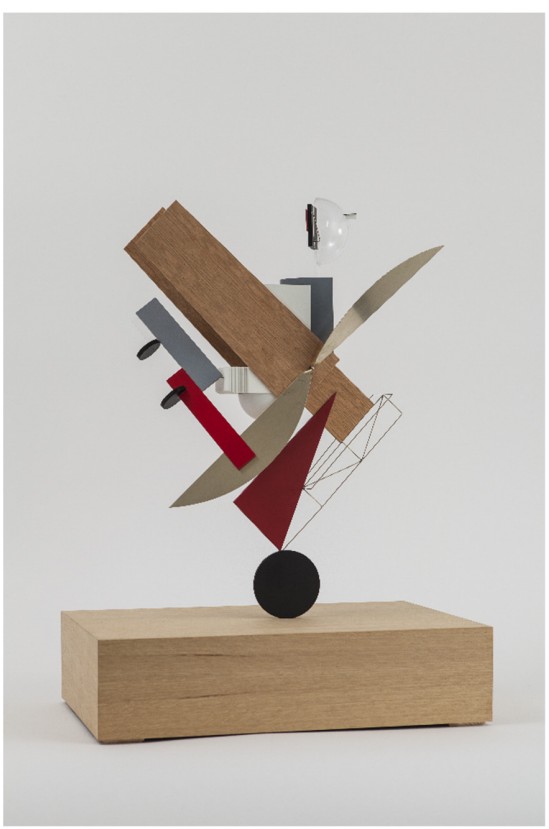

**Figure 18.** Model after El Lissitzky, 'The Time Traveller', 1923, construction by Henry Milner, 2009. Van Abbemuseum. Photo: Perry van Duijnhoven.

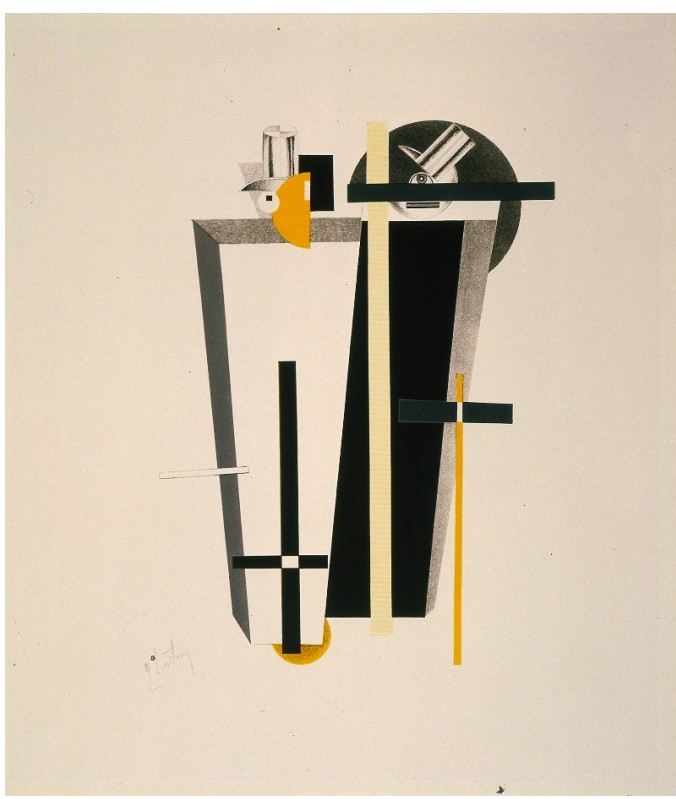

**Figure 19.** El Lissitzky, 'The Gravediggers, lithograph for the 'Figurinnenmappe', 1923. Van Abbemuseum. Photo: Peter Cox.

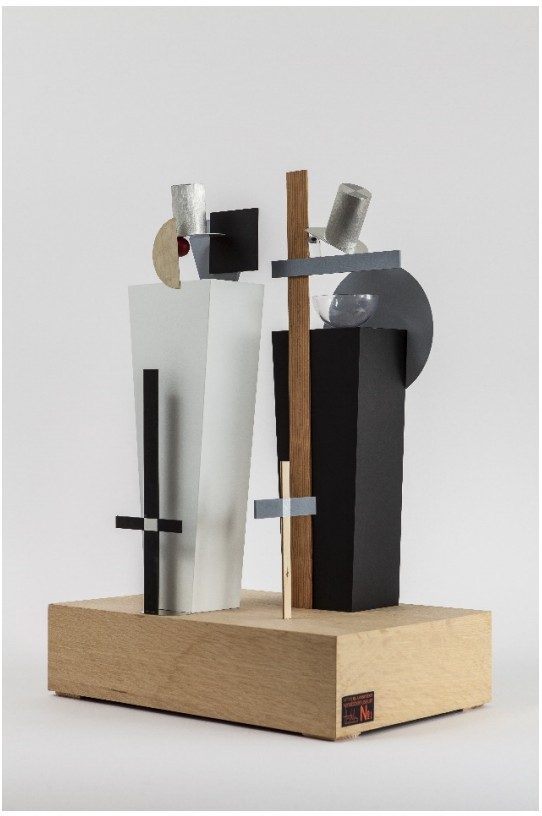

**Figure 20.** Model after El Lissitzky, 'The Gravediggers', 1923, construction by Henry Milner, 2009. Van Abbemuseum. Photo: Perry van Duijnhoven.

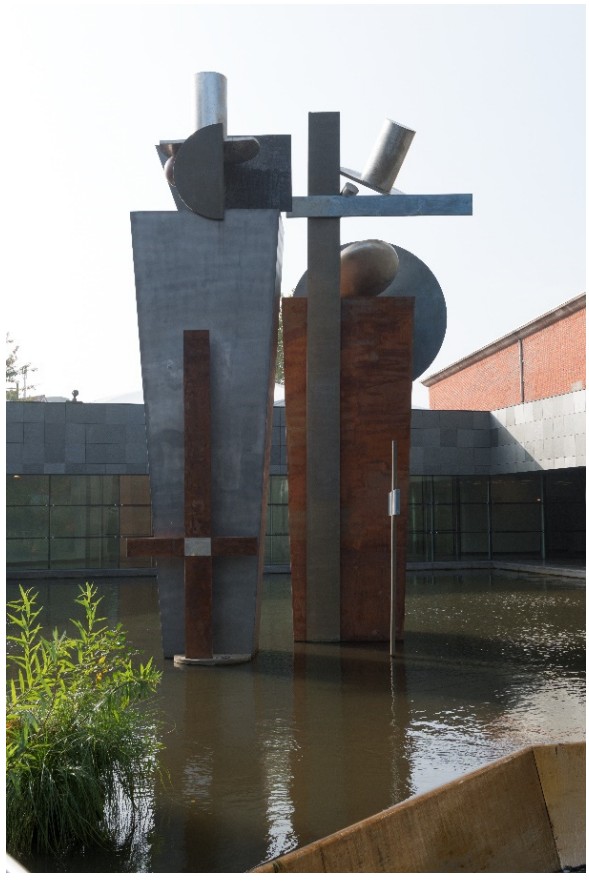

**Figure 21.** Statue after El Lissitzky, 'The Gravediggers', construction designed by Henry Milner, 2009. Museum Danubiana, Bratislava. Photo: Peter Cox.

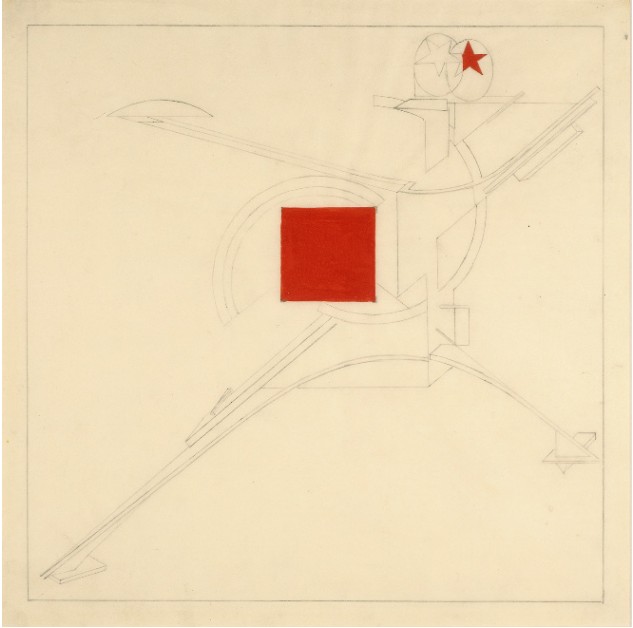

**Figure 22.** El Lissitzky, 'The New Man', drawing for the 'Figurinnenmappe', 1923. Van Abbemuseum. Photo: Peter Cox.

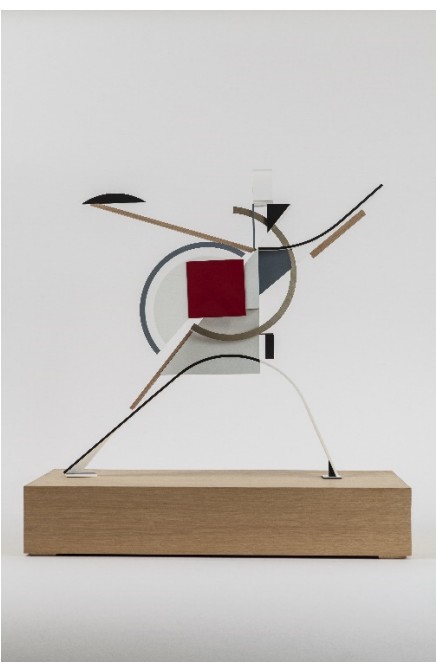

**Figure 23.** Model after El Lissitzky, 'The New Man', 1923, construction by Henry Milner, 2009. Van Abbemuseum. Photo: Perry van Duijnhoven.

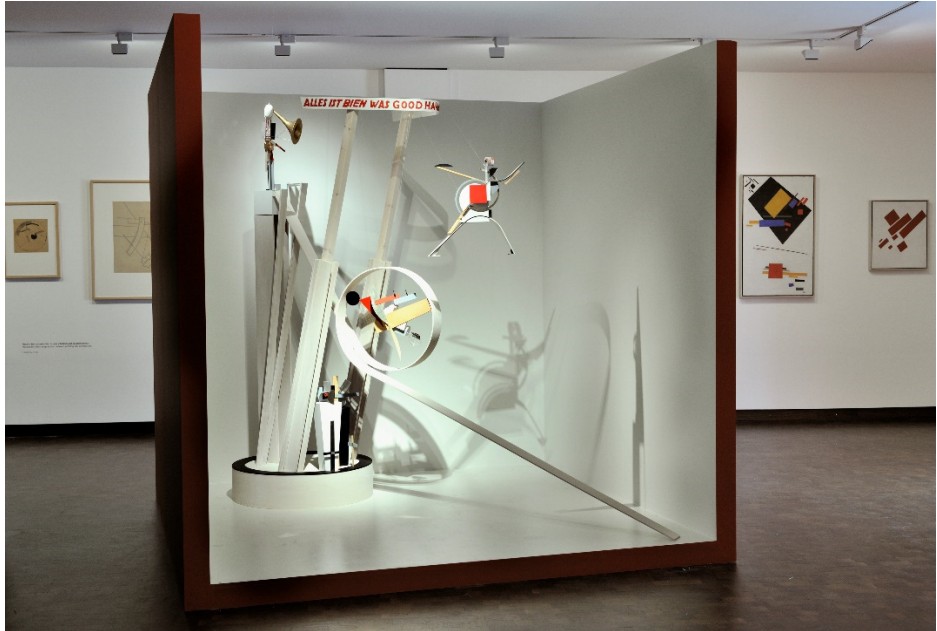

**Figure 24.** View of one of the rooms in the exhibition 'Lissitzky+: Victory over the Sun', Van Abbemuseum, 2009. Front: Model after El Lissitzky, 'The Viewing Machine', construction by Henry Milner, 2009. Van Abbemuseum. Photo: Peter Cox.

In this exhibition concept of spatiality, Lissitzky's architectural designs were essential. Although never actually built, these drawings were intended to be made into buildings. They were presented in a separate room of the exhibition, together with models of several designs by other artists and architects. The presentation in this space included a towering maquette of Lissitzky's 'Wolkenbügel', a skyscraper designed in 1924. Also on view was a model of 'Proun 1 E, The City' after a lithograph by Lissitzky of 1921 (Figure 25). An earlier version of Lissitzky's 'New Man', a lithograph from the so called 'Kestner Porfolio' made in

1923, was transformed into a blown-up hanging version and was mounted on the window in the stairwell (Figure 26).

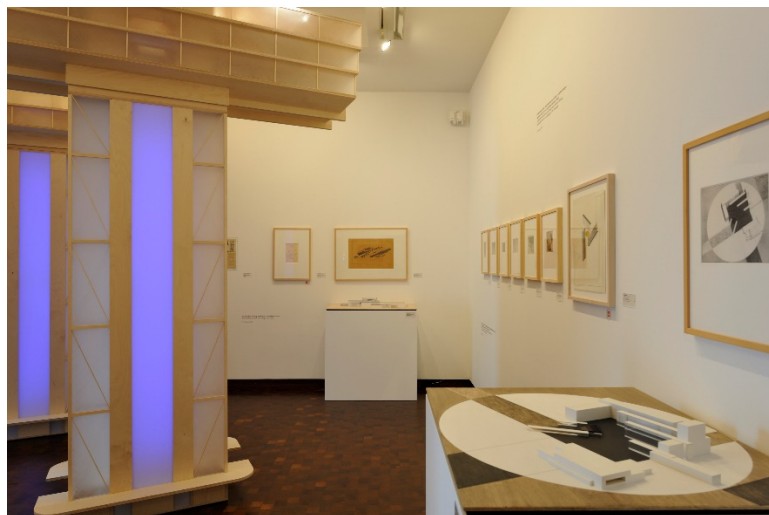

**Figure 25.** View of one of the rooms in the exhibition 'Lissitzky+: Victory over the Sun', Van Abbemuseum, 2009. Left: Model after El Lisstzky, 'Wolkenbügel', construction by Van Abbemuseum, 2009. Right: Model after El Lissitzky, 'The City', construction by Henry Milner, 2009. Van Abbemuseum. Photo: Peter Cox.

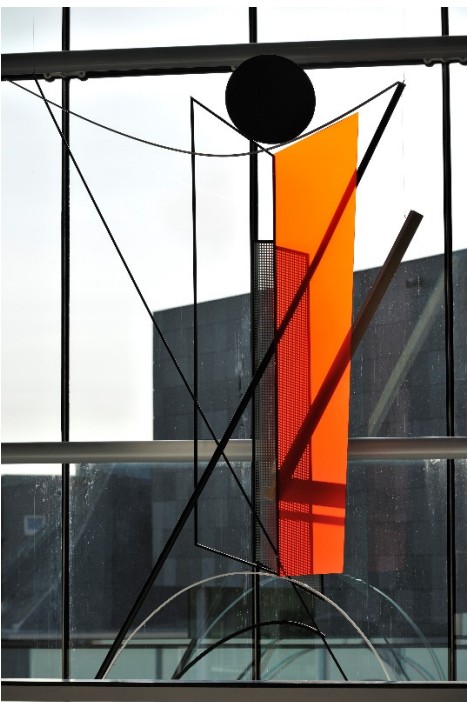

**Figure 26.** Model after El Lissitzky, 'The New Man', 1923, hanging construction by Henry Milner, 2009. Van Abbemuseum. Photo: Peter Cox.

The exhibition 'Lissitzky+: Victory over the Sun' opened in September 2009 and ran for one year. It was accompanied by an extensive educational program, a symposium and several other activities. Attracted by the positive reviews in the press, many visitors came to the museum. Taking designs by Lissitzky one step further proved a useful educational concept: it encouraged the public to look closer and to compare Lissitzky's designs with the models made based on them.

In retrospect however, we might ask ourselves—myself as part of the curatorial team included—if some of these constructions were not a step too far. With his figurines, Lissitzky intended to make a theatrical performance. Constructing them as puppets thus would be the right continuation of this idea. But turning one design into a colossal statue in the museum's pond now seems over the top to me. Today, more than ten years later, I also have my doubts about the large 'New Man' model hanging in front of the big museum window as a sign board. Was this in line with Lissitzky's intentions? To think big is not always good advice.

## 7. Lissitzky's Bright Future and the Reverse

A very elaborate artistic dialogue with the work of Lissitzky followed when in 2010 the Van Abbemuseum invited the artists Ilya and Emilia Kabakov to conceive an exhibition using the Lissitzky collection. For this presentation, Ilya Kabakov came up with an ambitious plan. The curatorial team consisting of the artists, director Charles Esche and me, was immediately convinced: we would arrange both oeuvres in eight opposing pairs according to the following themes:

| Lissitzky | Kabakov |
| --- | --- |
| THE ARTIST AS A REFORMER | THE ARTIST AS A REFLECTING CHARACTER |
| THE COSMOS | VOICES IN THE VOID |
| CLARITY OF FORMS | GARBAGE |
| VICTORY OVER THE EVERYDAY | EVERYDAY'S VICTORY |
| MEMORY: MONUMENT TO A LEADER | MEMORY: MONUMENT TO A TYRANT |
| TRANSFORMING LIFE | ESCAPING LIFE |
| TRUST IN THE NEW WORLD | UNREALIZED UTOPIA |
| THE BRIGHT FUTURE AHEAD | THE BRIGHT FUTURE BEHIND US |

Following this concept, each room in the exhibition had its own theme and was divided into two parts: one for Lissitzky and one for Kabakov. The themes were indicated with big upper case letters on the wall, red for Lissitzky, black for the Kabakovs. Thus, each room compared a different aspect of the oeuvre of both artists. The resulting exhibition, carefully designed and prepared by the Kabakovs in collaboration with the museum, took place on two floors in the new building of the Van Abbemuseum. It opened there in December 2012. The exhibition subsequently travelled to the Hermitage in St. Petersburg, the Multimedia Art Museum in Moscow and the Kunsthaus in Graz[20].

At first sight, the title of the exhibition, 'Lissitzky–Kabakov: Utopia and Reality', led many to believe that Lissitzky was the utopian and Kabakov the realist here. But a closer look at the presentation was enough to conclude that it was more complicated: aspects of both utopia and reality were present in the work of the two artists in totally different ways. And despite the many differences in their oeuvres there were also similarities to be discovered.

To make the intended juxtapositions visually more effective, some of the Kabakovs' installations were recreated, and some new Lissitzky models were needed as counterparts. Once again, the Van Abbemuseum asked Henry Milner to design and execute these. The division of each room into a Lissitzky and a Kabakov part worked very well. The visitors had to walk in between the two and involuntarily asked themselves which of these artistic visions on Soviet society was the right one.

Two examples will have to be sufficient to illustrate the exhibition concept here. In the room 'Victory over the Everyday' and the corresponding 'Everyday's Victory', the Lissitzky part was left almost empty. His 1930 floorplan for an apartment to solve the housing shortage was executed in lines on the floor to indicate the intended available space. A few historical photographs, a reconstruction of the Lissitzky chair of the same year (Figure 27)

and a reconstruction of the 1927 maquette for a communal house were all of the exhibits in this part of the room (Figure 28). The opposite part showed a plethora of Kabakov works that had to do with daily Soviet life in and around the communal kitchen. One wall was filled to the ceiling with paintings, and along the other two walls were parts of the 1991 'Communal Kitchen' installation (Figures 29 and 30).

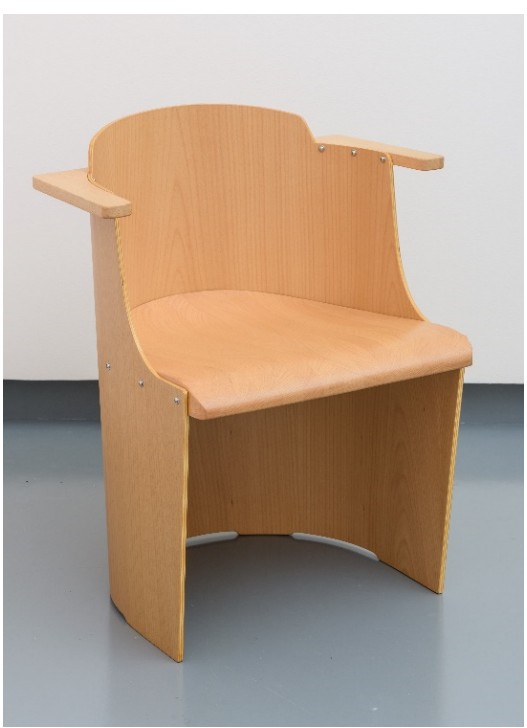

**Figure 27.** El Lissitzky, 'Armchair of bent plywood', 1930, reconstruction by Mark van den Heuvel, 2012. Van Abbemuseum. Photo: Peter Cox.

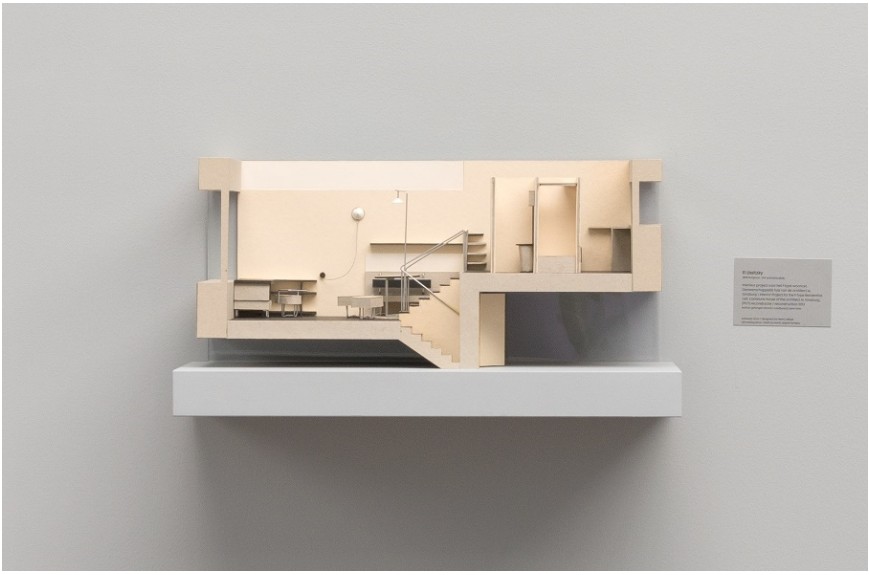

**Figure 28.** Model after El Lissitzky, Interior Project for the F-type Residential Cell, 1927, reconstruction by Henry Milner, 2012. Van Abbemuseum. Photo: Peter Cox.

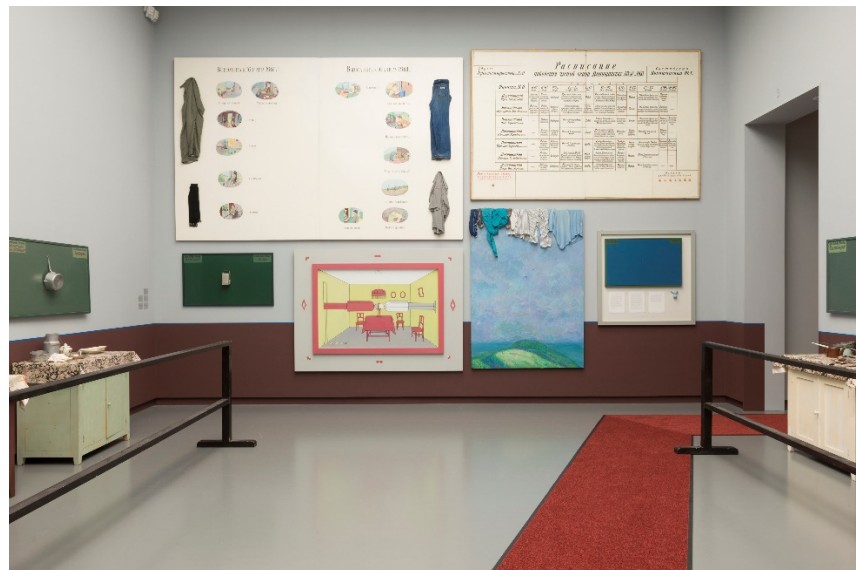

**Figure 29.** View of one of the rooms with works by Ilya Kabakov in the exhibition 'Lissitzky–Kabakov: Utopia and Reality', Van Abbemuseum, 2012. Photo: Peter Cox.

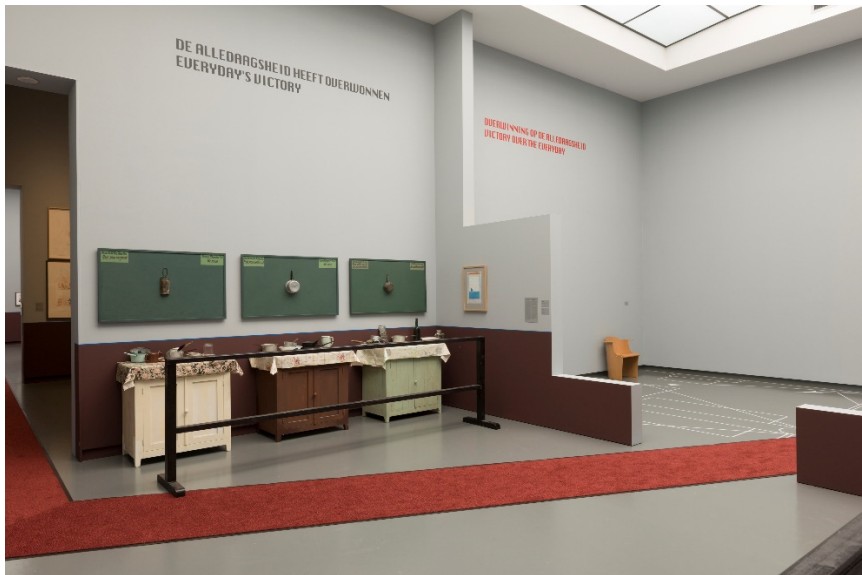

**Figure 30.** View of one of the rooms in the exhibition 'Lissitzky–Kabakov: Utopia and Reality', Van Abbemuseum, 2012. Front: part of the installation by Ilya Kabakov, 'In the Communal Kitchen', 1991. Back: El Lissitzky, 'Armchair of bent plywood', 1930, reconstruction by Mark van den Heuvel, 2012. On the floor at the back: El Lissitzky, Floorplan for an apartment to solve the housing shortage', 1930, executed in actual measurements by Van Abbemuseum, 2012. Photo: Peter Cox.

Another theme in the exhibition was propaganda. Lissitzky's part, 'The Bright Future is Ahead of Us', consisted of hardcore Soviet propaganda: photos and reconstructions of objects related to the 'Pressa' exhibition that was held in 1928 in Cologne. For the occasion, the grandiose 'Pressa Star' (Figure 31) was put up on one side of the room. The other part, 'The Bright Future Is Behind Us', showed a new version of the Kabakovs' agitprop propaganda wagon of 2004 (Figure 32). Inside you could hear women singing heroic Soviet songs.

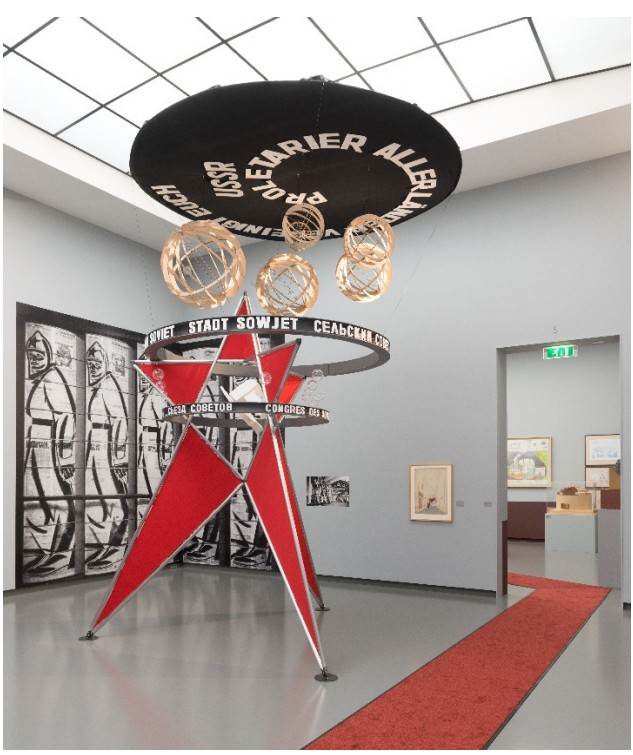

**Figure 31.** View of one of the rooms in the exhibition 'Lissitzky–Kabakov: Utopia and Reality', Van Abbemuseum, 2012. Front: model after El Lissitzky, 'Pressa Star', 1927, reconstruction by Henry Milner, 2012. Van Abbemuseum. Photo: Peter Cox.

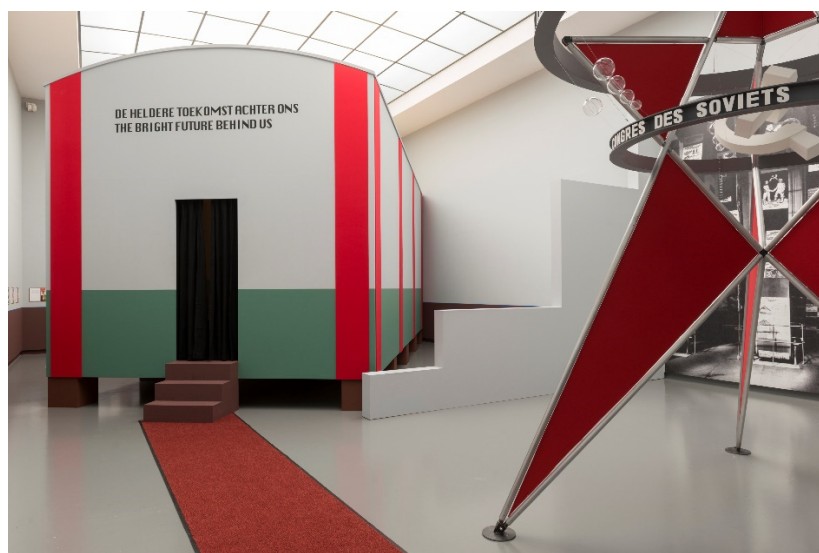

**Figure 32.** View of one of the rooms in the exhibition 'Lissitzky–Kabakov: Utopia and Reality', Van Abbemuseum, 2012. Front: part model after El Lissitzky, 'Pressa Star', 1927, reconstruction by Henry Milner, 2012. Van Abbemuseum. Back: Ilya Kabakov, 'Let's Go Girls', 2012. Van Abbemuseum. Photo: Peter Cox.

This exhibition was the first time that these two 20th-century Russian artists had been presented together. It completed a circle that started with the revolutions in the early years of the twentieth century and finished with the upheavals of 1989. This confrontation between early Soviet art and that of the late Soviet era did not consist solely of original artworks. It was largely made possible thanks to numerous meticulous reconstructions. In

this way, the exhibition enabled a broad public to obtain a better understanding of the art and culture of the intervening period.

One reason the Lissitzky reconstructions worked so well in this exhibition, was that they were shown together with some of the Kabakov models. For many of their projects, Ilya and Emilia Kabakov made prototypes. And like some of Lissitzky's projects, some of the Kabakov projects are still not realised. For both Lissitzky and the Kabakovs, utopia is a well-known place.

## 8. Lissitzky Even More Contemporary

Sarah Pierce (1968, Dublin, Ireland) was one of four artists participating in the exhibition 'Positions #2', curated by Annie Fletcher (Van Abbemuseum, 28 November 2015–3 April 2016)[21]. In this exhibition she showed three installations, one of which, 'Gag', made in 2015, was based on the work of El Lissitzky and Alice Milligan. Lissitzky needs no introduction in this context. Alice Milligan (1865–1953, Omagh, Ireland) was a poet and writer. As a nationalist she promoted Gaelic, the Irish language, and understood amateur theatre as a medium to involve rural communities in the Irish cultural revolution.

The starting point of the installation 'Gag' (Figure 33) was an 'archive of debris': in the lead up to Milligan's presentation the unused scraps and remnants from the last exhibition —plinths, presentation walls and other material normally thrown away—were amassed. In her resulting installation, these leftovers not only made the immediate past of these particular museum rooms visible and tangible, they also related visually to the photographs of works in the 'First Constructivist Exhibition' in 1921 as reconstructed in the Tretyakov Gallery and to a photograph of Malevich's 'Black Square'. On the walls, the 'Proun' lithographs from the Kestner portfolio mentioned earlier were displayed, including the isometric layout of the 'Proun' room.

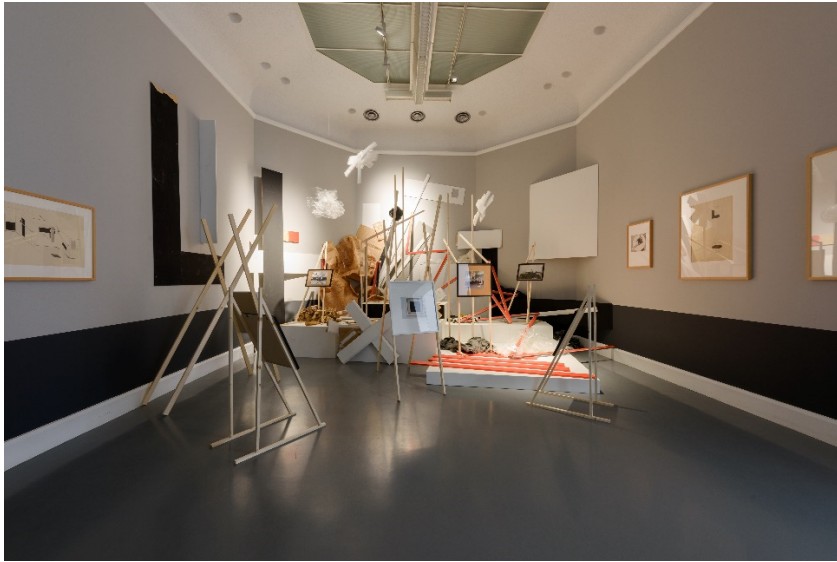

**Figure 33.** View of one of the rooms in the exhibition 'Positions #2', Van Abbemuseum, 2015. Sarah Pierce, 'Gag', 2015. Photo: Peter Cox.

Looking back at this project, Sarah Pierce says:

( . . . ) Lissitzky's work is important for my work, although I am not at all interested in either the cult of the artist or the cult of genius. In fact, my turn to Lissitzky has less to do with specific historical works created in his lifetime or the artist himself as a historical figure and more with what I read as the interplays between a specific and general address in his work. Lissitzky made his works in direct dialogue with artists around him. This transfer between artists, how one person's work changes through the work of others, has always interested me.

What also strikes me is that I can still take up this address a century later. This is important because it speaks to a way that artists work, really, across geographies and times. I do not mean this as a universal address—it is not received in the same way by everyone, or every artist. ( . . . ) In these projects my concern is how to take up the work of these artists, not as an act of reverence, but as a real and material transfer of concepts across time. The best art history understands how to do this. It requires a complete institutional shift, away from individuals, biographies and the intrigue that often compels historians to look closely at the figure. Instead, we should turn to the signs and symbols of a total system of art making. That is what we can get from artists such as Lissitzky. ( . . . ) I would like to think more about Lissitzky as a teacher. Did he teach? I have learned from him. I have looked at his contributions to the language of art and thought "Oh good, that is allowed. Then that is what I'll do!"

## 9. Lissitzky in Person

Over the years, the Van Abbemuseum has collected letters and other correspondence of Lissitzky. Many of these writings were directed to his colleagues and show Lissitzky as a serious and tireless worker on many different artistic projects. But occasionally, one gets a glimpse of his character, his intelligence and his sense of humor. A letter written to his wife Sophie while convalescing in Switzerland, for instance, includes a self-portrait of the artist working on some of his projects (Figure 34).

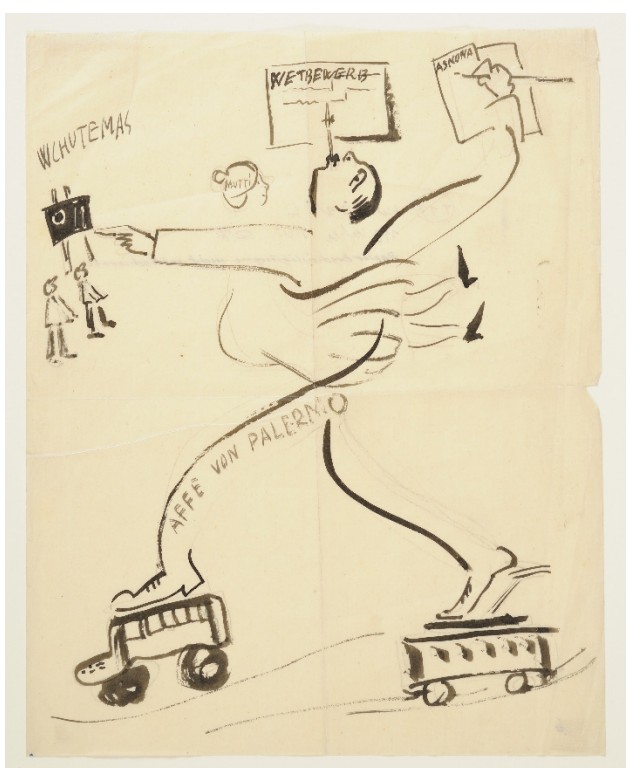

**Figure 34.** El Lissitzky, Drawing in a letter to Sophie Küppers, 1924. Van Abbemuseum. Photo: Peter Cox.

In 2017, on the occasion of the 100th anniversary of the Dutch artistic movement De Stijl, curator Diana Franssen and librarian Willem Smit organised an exhibition in the museum library on the correspondence of Lissitzky with his Dutch colleague and friend, the architect J.J.P. Oud (Figure 35). This correspondence, acquired in 1970, is archived in the Van Abbemuseum and only accessible for research purposes. The written texts in German were now transcribed and translated into Dutch in a publication (Franssen 2017).

This exchange of letters and postcards, which took place between 1923 and 1928, not only enables us to follow Lissitzky on his tour through Europe but also to see the two friends' opinions on the situation in art and politics at the time and how they developed and discussed artistic theories.

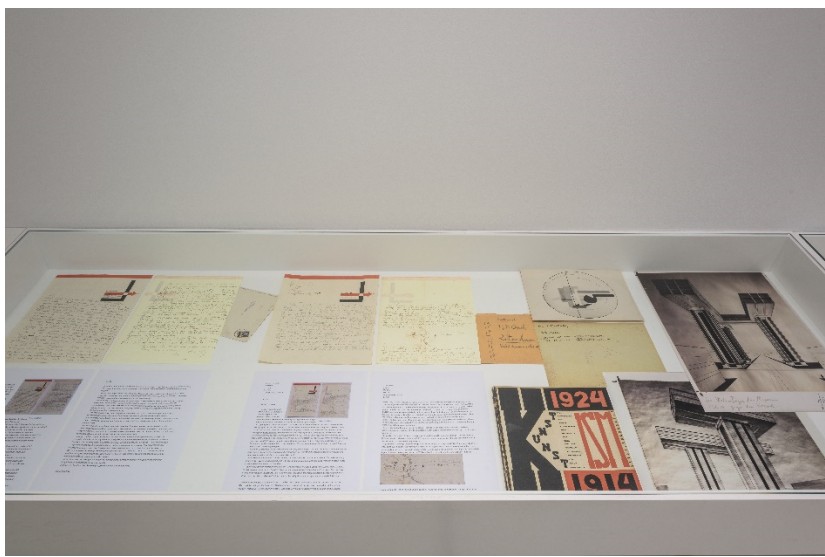

**Figure 35.** View of one of the vitrines in the exhibition '100 Years De Stijl: Lissitzky and Oud in dialogue', Van Abbemuseum, 2017.

## 10. A Collection Traveling around EUROPE

In recent years, several opportunities arose to show large parts of the Lissitzky collection outside the Van Abbemuseum, either in solo exhibitions or in combination with the work of other artists. In 2015, a large part of the collection travelled to the Irish Museum of Modern Art (IMMA) in Dublin as part of the exhibition 'El Lissitzky: The Artist and the State'. Lissitzky's works were shown there alongside archival material related to Alice Milligan and newly commissioned and recent works by several contemporary artists. Two years later, many works were lent to the major Lissitzky retrospective in the Tretyakov Gallery and the Jewish Museum and Tolerance Centre in Moscow, and in 2018 many works from the Lissitzky collection were on show in the exhibition 'Chagall, Lissitzky, Malévitch–l'Avant-garde Russe à Vitebsk 1918–1922' in the Centre Pompidou in Paris. In 2019 most of the Van Abbemuseum's Lissitzky collection was shown in the Danubiana Museum in Bratislava.

## 11. Epilogue

Ever since Lissitzky came to Eindhoven, his works have been an inspiration for the Van Abbemuseum and its public. The fact that Lissitzky was very open to the art of Western Europe and was in dialogue with many of his colleagues there made his entrance into the collection a kind of homecoming. Forming a focus there from the beginning in 1965, the open character of Lissitzky's oeuvre and the fact that others can participate and even finish it appealed to several of the museum's directors and curators and to many visitors.

In retrospect, not every way in which the museum dealt with this artistic heritage was successful. A few of the Lissitzky reconstructions were over the top and some of the artists' projects with his work were more successful than others. Art historians often meticulously reconstruct the personal history and the oeuvre of an artist. But other artists can take the liberty to use (art)history as a flexible and dynamic tool to create new works. Looking at such works, we may realise that we too can use art as a source of inspiration.

For more than one hundred years, Lissitzky's works have been aiming at the future with the intention to be of use. It is this aiming at a new reality that gives many of his artworks an unfinished character. They invite us to work further in this direction. Because

of his pluriform and open way of creating, Lissitzky is one of those artists who allows such dynamic use. In these cases, a museum can help to enforce this creative impulse and pass it on to the public.

**Funding:** This research received no external funding.

**Data Availability Statement:** Archival material on all exhibitions of the Van Abbemuseum can be found at: https://vanabbe.inforlibraries.com/abbeweb/Vubis.csp?Profile=Default, accessed on 11 October 2022.

**Conflicts of Interest:** The author declares no conflict of interest.

## Notes

1    For this article I have used several unpublished lectures that I gave in Moscow, Novosibirsk, Belgrade and Cambridge. I also used one of my published articles: (Renders 2017). I would like to thank Ruth Addison for editing and improving my English text.

2    For a survey of Jean Leering's directorship of the Van Abbemuseum, see: (Pingen 2005, ch. 3).

3    (Lissitzky-Küppers 1967). For the English version, see: (Lissitzky-Küppers 1980).

4    For a more detailed account of the acquisition of the Lissitzky collection by the Van Abbemuseum and the contacts between Klihm and Leering, see: (Pingen 2005, pp. 224–26).

5    For the text, music and designs for this opera as well as numerous essays on the different aspects of this work, see: (Railing 2009).

6    "Die weitere Bearbeitung und Anwendung der hier niedergelegten Ideen und Formen überlasse Ich den Anderen und gehe selbst an meine nächste Aufgabe" (my English translation).

7    "Anleitung zum Handeln" (my English translation). Mentioned in: (Kempers 2018, p. 67).

8    For an introduction to the 'Prouns', see: (Goryacheva 2017). In 1924, Lissitzky made an inventory of his 'Prouns', see: (Nisbet 1987, pp. 155–76). Perloff (2003) offers a chronological overview and an analysis of the artistic legacy of Lissitzky.

9    Lissitzky published his design for the Prounenraum in the first issue of the magazine *G* in 1923. For the text of this publication, see: (Leering et al. 1965, p. 57).

10    (Berndes 1999, pp. 62, 63). "Ik kreeg een prent onder ogen van de Prounenraum die in isometrie was getekend. Door de deurhoogte te meten en er van uit te gaan dat de standaardhoogte van een deur twee meter tien is, kwam ik erachter dat de prent op schaal 1: 20 was uitgevoerd. Toen werd het duidelijk dat we de Prounenraum konden reconstrueren." (my English translation).

11    (Berndes 1999, p. 67). "Een reconstructie heeft een andere status dan een origineel, dat moge duidelijk zijn, ook al wordt het origineel zeer dicht benaderd. Want het is de bedoeling met een reconstructie een ervaring op te roepen die nagenoeg gelijk is aan de ervaring van het origineel, dat niet meer in de originele staat aanwezig is. Ook in het geval van reconstructies ging het me om het stimuleren van het beeldvormingsvermogen van het publiek." (my English translation).

12    (Leering et al. 1965, p. 14): "Meiner Ansicht nach bedeutete das Gestalten von Demonstrationsraüme für Lissitzky eine Weise der schöpferischen Betätigung, worin sich diese Polarität auflöste. Hier könnte er >Gegenstände< schaffen, von konkreter Art, mit geistiger Aussagekraft, die sich weit über die von Industrial Design erhob. So wie bei seiner Typografie ging er auch hier davon aus, daß das Gezeigte auf der Zuschauer so übertragen werden mußte, daß sie in die Übertragung einbezogen wurden, d.h. die Kreativität der Zuschauer selbst frei gemacht wurde." (my English translation).

13    Leering gave this lecture at the end of the exhibition, on January 10 1966 in the Van Abbemuseum. This quote is from: (Kempers 2018, p. 68). "In beider conceptie treedt beweging op, maar bij Malewitch is het meer alsof hij de beschouwer een beweging in de ruimte laat zien. Lissitzky doet de beschouwer in zijn beschouwing zelf bewegen in die ruimte. Dit laatste staat centraal in het werk van Lissitzky, en dit principe: de beschouwer actief in de beelding betrekken, is ook uitgangspunt voor zijn ander werk, zoals zijn typografie en demonstratieruimten ( . . . )" (my English translation).

14    Cited in: (Pingen 2005, p. 226). "De Eindhovense tentoonstelling is allerminst een eindpunt: waar het Lissitzky om ging staat nog te gebeuren. Inspirerender kan een tentoonstelling nauwelijks zijn." (my English translation).

15    For a more detailed account of the ups and downs of the purchasing process, see: (Pingen 2005, pp. 262, 263).

16    For more information on the preparations and the exhibition tour, see: (Pingen 2005, pp. 483, 484).

17    For the comment in the catalogue, see: (Debbaut 1990, p. 217). For the claim see: (Pingen 2005, p. 484).

18    For the history of the Hanover reconstructions of the Abstract Cabinet, see: (Krempel 2015).

19    For more information on this project and the interview with Deimantes Narkevicius, see: (Renders 2017, pp. 94–96).

20    The exhibition was accompanied by a catalogue in Dutch, English and Russian: (Renders 2012; Дианова 2013). For more information on this exhibition, see: (Renders 2017, pp. 96–99).

21    For more information on this project and the interview with Sarah Pierce, see: (Renders 2017, pp. 100–2).

# References

## Exhibitions Discussed

1965, 1966: *El Lissitzky*, Van Abbemuseum, Eindhoven, Kunsthalle Basel, Kestner-Gesellschaft, Hanover.

1990, 1991: *El Lissitzky: 1890–1941: Architect, Painter, Photographer, Typographer*, Tretyakov Gallery, Moscow, Van Abbemuseum, Eindhoven, Fundaçion Caja de Pensiones, Madrid, Musée d'art moderne de la ville de Paris.

2006: *Plug In #6*, Van Abbemuseum, Eindhoven.

2009, 2010: *Lissitzky+: Victory over the Sun*, Van Abbemuseum, Eindhoven.

2012, 2013, 2014: *El Lissitzky, Ilya/Emilia Kabakov: Utopia and Reality,* Van Abbemuseum, Eindhoven, Hermitage, St. Petersburg, Multimedia Art Museum, Moscow, Kunsthaus, Graz.

2015: *El Lissitzky: The Artist and the State*, Irish Museum of Modern Art (IMMA), Dublin.

2015, 2016: *Positions #2*, Van Abbemuseum, Eindhoven.

2017: *El Lissitzky*, Tretyakov Gallery and the Jewish Museum and Tolerance Centre, Moscow.

2017: *Lissitzky en Oud in dialoog*, Van Abbemuseum, Eindhoven.

2018: *Chagall, Lissitzky, Malévitch—l'Avant-garde Russe à Vitebsk 1918–1922*, Centre Pompidou, Paris.

2019: *El Lissitzky*, Danubiana Museum, Bratislava.

## Secondary Sources

Berndes, Christiane, ed. 1999. *Een collectie is ook maar een mens: Edy de Wilde, Jean Leering, Rudi Fuchs, Jan Debbaut over verzamelen*. Eindhoven Rotterdam: Stedelijk Van Abbemuseum NAi.

Debbaut, Jan, ed. 1990. *El Lissitzky: 1890–1941: Architect, Painter, Photographer, Typographer*. Eindhoven: Van Abbemuseum.

Franssen, Diana, ed. 2017. *100 Jaar De Stijl: Lissitzky en Oud in dialoog*. Eindhoven: Van Abbemuseum.

Goryacheva, Tatyana. 2017. The Proun Station. In *El Lissitzky*. Edited by Tatyana Goryacheva. Moscow: Jewish Museum and Tolerance Center & State Tretyakov Gallery, pp. 30–49.

Kempers, Paul. 2018. *'Het gaat om heel eenvoudige dingen': Jean Leering en de kunst*. Amsterdam: Valiz.

Krempel, Ulrich. 2015. Kurt Schwitters' Merzbau und El Lissitzkys Kabinett der Abstrakten. Zwei Rekonstruktionen von zerstörten Räumen der Moderne im Sprengel Museum Hannover. In *Die Austellungskopie. Mediales Konstrukt, Materielle Rekonstruktion, historische Dekonstruktion*. Edited by Annette Tietenberg. Köln: Böhlau Köln, pp. 115–28.

Leering, Jean, Joost Baljeu, Camilla Gray, Mart Stam, Dietrich Helms, Schuldt, and El Lissitzky. 1965. *El Lissitzky*. Eindhoven: Stedelijk van Abbemuseum.

Lissitzky, El, and Jean Arp. 1925. *Die Kunstismen = Les Ismes De L'art = the Isms of Art*. Erlenbach-Zürich: Eugen Rentsch.

Lissitzky, El, and Kasimir Malevich. 1922. *Pro Dva Kvadrata: Suprematicheskiĭ Skaz: V 6-Ti Postroĭkakh. [About Two Squares.]*. Berlin: Skify.

Lissitzky-Küppers, Sophie. 1967. *El Lissitzky: Maler, Architekt, Typograf, Fotograf*. Dresden: Verlag der Kunst.

Lissitzky-Küppers, Sophie. 1980. *El Lissitzky, Life, Letters, Texts*. London: Thames and Hudson.

Nisbet, Peter. 1987. *El Lissitzky, 1890–1941: Catalogue for an Exhibition of Selected Works from North American Collections, the Sprengel Museum Hanover, and the Staatliche Galerie Moritzburg Halle*. Cambridge: Harvard University Art Museums, Busch-Reisinger Museum.

Pingen, René. 2005. *Dat museum is een mijnheer: De geschiedenis van het Van Abbemuseum 1936–2003*. Amsterdam: Artimo.

Perloff, Nancy, ed. 2003. *Situating El Lissitzky: Vitebsk, Berlin, Moscow. Issues & Debates*. Los Angeles: Getty Research Institute.

Railing, Patricia. 2009. *Victory over the Sun*. Forest Row: Artists Bookworks.

Renders, Willem Jan, ed. 2012. *El Lissitzky, Ilya/Emilia Kabakov: Utopia and Reality*. Eindhoven: Van Abbemuseum.

Renders, Willem Jan. 2017. Why Lissitzky? In *El Lissitzky*. Edited by Tatyana Goryacheva. Moscow: Jewish Museum and Tolerance Center & State Tretyakov Gallery, pp. 94–102.

Дианова, Елена, ед. 2013. Эль Лисицкий, Илья и Эмилия Кабаковы: Утопия и Реальность. Санкт-Петербург: Государтственный Эрмитаж.