# Peer review of "“You Can Do This”: Working with the Artistic Legacy of El Lissitzky"

_arts_

Round 1
Reviewer 1 Report
This is a very interesting article that presents significant material clearly and sensibly, demonstrating emphatically the role that the Van Abbemuseum has played in establishing the historical importance of El Lissitzky and his relevance to contemporary art.
For me, it particularly stimulates reflection on institutional policy in museums - notably smaller regional ones; on the impact of specific exhibitions on the art market (and I was struck by Jan Leering's astute tactics in this respect); and on the role that reconstructions of lost works can play in exhibitions
expression/typos:
lines 60-61 - it might be clearer if put as 'first right to buy'.
line 169 insert: 'for the catalogue'; line 591 insert 'sense of humor'
Author Response
Please find enclosed the final version of my article. I have changed the original text according to most of the remarks of the peer reviewers, I have added a list of exhibitions and the text has been edited by a native speaker of English.
Reviewer 2 Report
The article is very informative and didactic and I did enjoy it. In our field: the Russian avant-garde it is very appropriate today to reconsider how we, as critics, as historians, approached the subject. i.e., to create a new history of the history.
For this reason, I would had appreciated a schematic synopsis of the sequence of exhibitions for teaching or research purposes.
Author Response

(The authors gave the same response as above.)

Reviewer 3 Report
I understand the article as a kind of "collection history", which is very interesting. However, I miss the critical approach/a scientific question or something like an argumentation. For what purpose is the chronology of the "collection history" presented? There should be a clear research question at the beginning and a critical evaluation at the end. For example, has the museum dealt appropriately with the open nature of El Lissitzky's work, etc.? The arguments are implicitly there but not explicitly stated. For this reason, the essay is more of a presentation than a discussion in which the arguments are weighed against each other. Once arguments, conclusions, and evaluations are more clearly worked out, the essay will be very good.
Author Response

(The authors gave the same response as above.)

Reviewer 4 Report
This article presents a fascinating history of the collection of works by El Lissitzky in the Van Abbemuseum, Eindhoven: the aquistion of various items at different times; the numerous exhibitions of that material and the underlying intentions of the displays; the reconstructions of various projects and the construction of items that were not constructed in the artist's lifetime and were perhaps not intended for construction on the scale at which they have been realized; and the way in which Lissitzky's works have inspired other artists, leading to exhibitions at the museum such as that juxtaposing the work of Lissitzky and Kabakov.
Of course, much of this information is common knowledge; catalogues, reviews, etc are in the public arena. Even so, the author manages to add some new details and bring all that information together. The article thus amplifies existing knowledge. It ldoes not reveal totally new perspectives on the artist or the museum, but it is a useful history to have in the public realm, and should clearly be published by Arts.
That said, it should be stressed that there are several problems with the text at present.
Its tone is laudatory and tends to be uncritical of the policies adopted or the activities organized by the museum. This gives the article the character of a museum press release and conveys the impression that the article was written by an employee of the museum. Likewise the stress is on the various directors, while the curators and the curatorial team are not mentioned. Lissitzky's works were clearly used as a springboard for other creative actions and events, many of which were highly laudable, but the author seems to find no problem at all in the fact that drawings in a portfolio were then constructed as large sculptures.
Unfortunately, the congratulatory tone extends to the discussion of the exhibitions involving contemporary artists working in tandem with Lissitzky's works. These events could and should be dealt with in a more analytical and critical way. It is difficult to believe that there was no adverse criticism in the reviews of these shows, some of which seem more than slightly controversial. The author clearly wishes to reveal the relevance of Lissitkzky's work for contemporary artists - but this section is longwinded and far too narrative. I would advise dealing with it in a more thematic way and shortening this section.
In addition to modifying these parts of the article, the author should also expand the bibliography. At present, there is little in English. Yet the literature on Lissitzky in the English language is extensive. The Sophie Lissitzky-Kuppers book was translated into English and this edition should be cited. Likewise Peter Nisbet's exhibition in Boston should be mentioned.
The author's English is on the whole good, but it is a long text and there are numerous points that could and should be adjusted. Egs
line 25 - Lissitzky was Jewish. Today problems of nationality and ethnicity are coming to the fore. Lissitzky was Jewish but a citizen of the Russian Empire. It might be best to describe him simply as an avant-garde artist
27 insert 'the' before 'Bauhaus'
35 'for his', not 'of his exhibition'
38 and 45 'people'not 'persons'
39 hyphen needed - 'well-known'
40 'film maker' not 'filmer'
45 'gallery owner', not 'gallerist'
50 'at' not 'in'
51 'extended convalescence' not 'a long cure'
54 change to read 'asked for them back'
55 'to' not 'for'
59 'watercolors'
60 rephrase 'obtained first refusal on purchasing the collection'
67 reference for quotation needed
68 change 'Russian avant-garde'
rephrase 'Despite Moscow's refusal to lend works'
75 'of' not 'in'
76 'reminiscences' not 'memories'. Memoirs are more extensive than reminiscences
79 'who'not 'that'
112 Fundgrabe is not a well-known German phrase, it would be more appropriate to use the English 'treasure trove'
112 'especially' not 'particularly'
115 'Proekt' not 'Projekt'
116 'represents' better here than 'is'
116 'which'not 'that'
125 'portfolio'not 'folder'
Note 9 change 'on' to 'to'
Note 10 the journal is G, No. 1, not G1
153 'at the Hayward Gallery'
156-161 - should be re-translated. This doesn't read well at all in English at the moment
166 'breadth'not 'breath'
169 't the catalogue' not 'for catalogue'
173 insert hypen 'so-called'
195 'improving the world by integrating art and life' is better than 'improvements to the world...'etc
200 'Western' not 'West'
200-1 'fully revealed' might be better than 'perceived'
209 'time preparing for the ...', not ' of preparations for the exhibition'
216 'Committee' not 'commission'
225 'complete', not 'finish'
273 correct spelling of names: Yve-Alain; Khan-Magomedov
386-394 - there is repetition here that should be eliminated or at least reduced
293 'Nazis' not 'Nazi's'
505 'Petersburger hanging of paintings' - rephrase
542 'late' not 'last'
627 tenses 'have been aiming at the future'
Author Response

(The authors gave the same response as above.)
